# Marine liquid cloud geometric thickness retrieved from OCO-2's oxygen A-band spectrometer

Mark Richardson[1,2], Jussi Leinonen[1], Heather Q. Cronk[3], James McDuffie[1], Matthew D. Lebsock[1], Graeme L. Stephens[1,4]

[1]Jet Propulsion Laboratory, California Institute of Technology, Pasadena, CA 91125, USA
[2]Joint Institute for Regional Earth System Science and Engineering, University of California Los Angeles, Los Angeles, CA CA 90095, USA
[3]Cooperative Institute for Research in the Atmosphere, Colorado State University, Fort Collins, CO 80521, USA
[4]Department of Meteorology, University of Reading, Reading, RG6 6BB, UK

*Correspondence to*: Mark Richardson (markr@jpl.caltech.edu)

**Abstract.** This paper introduces the OCO2CLD-LIDAR-AUX product, which uses the Cloud-Aerosol Lidar and Infrared Pathfinder Satellite Observation (CALIPSO) lidar and the Orbiting Carbon Observatory-2 (OCO-2) hyperspectral A-band spectrometer. CALIPSO provides a prior cloud top pressure ($P_{top}$) for an OCO-2 based retrieval of cloud optical depth, $P_{top}$ and cloud geometric thickness expressed in hPa. Measurements are of single-layer liquid clouds over oceans from September 2014 to December 2016 when collocated data are available. Retrieval performance is best for solar zenith angle < 45° and when the cloud phase classification, which also uses OCO-2's weak $CO_2$ band, is more confident. The highest quality optical depth retrievals agree with those from the Moderate Resolution Imaging Spectroradiometer (MODIS) with discrepancies smaller than the MODIS-reported uncertainty. Retrieved thicknesses are consistent with a substantially subadiabatic structure over marine stratocumulus regions, in which extinction is weighted towards the cloud top. Cloud top pressure in these clouds shows a 4 hPa bias compared with CALIPSO which we attribute mainly to the assumed vertical structure of cloud extinction after showing little sensitivity to the presence of CALIPSO-identified aerosol layers or assumed cloud droplet effective radius. This is the first case of success in obtaining internal cloud structure from hyperspectral A-band measurements and exploits otherwise unused OCO-2 data. This retrieval approach should provide additional constraints on satellite-based estimates of cloud droplet number concentration from visible imagery, which rely on parameterization of the cloud thickness.

## 1 Introduction

The Orbiting Carbon Observatory-2's (OCO-2) primary mission is to retrieve atmospheric $CO_2$ concentration (XCO2) using reflected sunlight (Crisp, 2015; Crisp et al., 2004; Eldering et al., 2016). This requires measurements at high spatial and spectral resolution with excellent signal-to-noise ratio, and OCO-2 measures spectra not just in the weak and strong CO2 bands, but also in the molecular oxygen (O₂) A-band near $\lambda = 0.78$ $\mu$m. The XCO2 retrieval is designed for clear skies and the A-band helps to identify and exclude cloudy scenes, and as of December 2016, between 88—93 % of soundings were not

used (Crisp et al., 2016). These soundings are rich in cloud information (Richardson and Stephens, 2018) and here we present the OCO2CLD-LIDAR-AUX product which exploits these unused OCO-2 data in concert with collocated measurements from the Cloud-Aerosol Lidar with Orthogonal Polarization (CALIOP) on the Cloud-Aerosol Lidar and Infrared Pathfinder Satellite Observations (CALIPSO) satellite.

For single layer marine clouds we retrieve cloud optical depth ($\tau$), cloud top pressure ($P_{top}$) and the geometric thickness ($H$), but express this $H$ in terms of cloud pressure thickness in hPa ($\Delta P_c$). $\Delta P_c$ is poorly constrained by other satellite products. For example, current spaceborne radar has insufficient sensitivity and range resolution to estimate the thickness of thin clouds while lidar is readily attenuated before reaching the cloud base in even moderately optically thick clouds. The cloud thickness has first-order relationship to liquid water path (LWP) and mixing between the boundary layer and free atmosphere

(Boers and Mitchell, 1994). Satellite estimates of the cloud droplet number density make assumptions about cloud vertical structure that parameterize the cloud geometrical thickness (see e.g. Grosvenor et al. (2018) for a recent summary). Realistic representation of the cloud droplet number is central to accurate representation of aerosol indirect effects in models (Jones et al., 1994; Lohmann and Feichter, 2005). Marine boundary layer clouds tend to have high albedo and warm tops, making them effective radiative coolers, and their response to current human-forced climate change is a major uncertainty in the

amount of global warming that will occur (Bony and Dufresne, 2005). Improved observational constraints on their properties will help modellers to improve the fidelity of their simulations and reduce uncertainty in projected climate change.

The ocean stratocumulus decks are a major contributor to low cloud radiative effects. One model study suggests that a global cooling of -8.0±0.1 W m$^{-2}$ could be achieved primarily by brightening these clouds through increasing cloud condensation nuclei (Latham et al., 2008). This is approximately the heating that would result from a quadrupling of atmospheric $CO_2$, and

such large potential radiative changes make understanding their processes very desirable. They tend to form where cool ocean water upwells near the western coasts of continents, particularly near California, Peru, Namibia and Australia. Unlike many other convective clouds, these are driven by cooling at cloud top rather than warming from the surface and a detailed summary of the processes involved can be found in Wood (2012).

The OCO2CLD-LIDAR-AUX product provides new information about low marine clouds, both for the OCO-2 native

footprints and collocated with CloudSat, allowing quick comparison with CloudSat radar and CALIPSO lidar cloud products. OCO2CLD-LIDAR-AUX is an iterative optimal estimation retrieval that uses a radiative transfer model to best fit a set of cloud properties to the observed spectrum. This paper describes the retrieval algorithm, data sources and modelling techniques; describes and validates outputs against other satellite products; and summarises and maps the retrieved cloud properties. It is organised as follows: Section 2 discusses the structure of the targeted clouds and the history and principle of

their retrieval; Section 3 describes the OCO-2 mission, instrumentation and orbit; Section 4 describes the retrieval; Section 5 explores the data with comparison to its priors, MODIS and CALIPSO; Section 6 reports and maps the full dataset retrieval statistics and Section 7 concludes.

## 2 Marine boundary layer clouds and retrieval of their geometric thickness

### 2.1 The subadiabatic cloud model

A common assumption for marine boundary layer clouds is that they follow an adiabatic or subadiabatic vertical profile, and clouds matching these assumptions have been observed in aircraft campaigns such as those partaking in the Aerosol Characterisation Experiment-2 (Pawlowska and Brenguier, 2000; Raes et al., 2000), and the predicted relationships between properties such as geometric thickness and liquid water path have also been measured (Painemal et al., 2017; Zuidema et al., 2012).

In this model, droplets in an air parcel are activated at the lifting condensation level (LCL) and as the parcel drifts upward, excess vapour condenses onto the droplets. In adiabatic conditions, liquid water content (LWC) is equivalent to the difference between the local saturation vapour pressure $e_s(z)$ and the saturation vapour pressure at the LCL. For the temperatures, pressures and relatively short altitude ranges of these clouds, this is well modelled by a linear LWC increase with height at a rate determined by the adiabatic condensation coefficient $c_w$. This is sometimes labelled $\Gamma_w$ and referred to as a lapse rate.

Non-adiabatic processes such as drizzle or entrainment of free tropospheric air at cloud top can change these profiles, resulting in sub-adiabatic conditions which can be parameterised (Betts, 1985; Boers and Mitchell, 1994). A set of assumptions is now commonly used, including constant cloud average values for sub-adiabiticity and the ratio between volume mean and effective droplet radius (e.g. Szczodrak et al. (2001)). Grosvenor et al. (2018) summarise many of the key relationships in a review of droplet number density, $N_d$, and derivations relevant for this study are in Supplementary Section 1.

In the subadiabatic model, geometric thickness is related to cloud-top effective radius $r_{e,ad,t}$ and cloud optical depth $\tau_c$ as:

$$H = \sqrt{\frac{20\rho_w r_{e,ad,t}\tau}{9Q_{ext}c_w f_{ad}}} \tag{1}$$

Where $\rho_w$ is the density of water, $Q_{ext}$ is the extinction efficiency ($Q_{ext}\approx 2$ for water droplets in the A-band) and $f_{ad} = 1$ in adiabatic conditions, and decreases with increasing subadiabaticity. We use this equation to derive the prior $H$ in OCO2CLD-LIDAR-AUX.

While real marine clouds in stable boundary layers can be well modelled by a subadiabatic structure, it is common for cloudy radiative transfer to assume plane-parallel clouds that are both horizontally and vertically homogeneous. For a fixed $LWP$ and $H$, a vertically homogeneous cloud has the same optical depth as a subadiabatic cloud provided that:

$$r_{e,h} = \frac{5}{6}r_{e,ad,t} \tag{2}$$

Where $r_{e,h}$ is the homogeneous cloud effective and this was derived and tested using cloud structures discretised at 25 m in the vertical in Brenguier et al. (2000). We use this argument to justify the combination of a subadiabatic cloud structure to derive prior $H$ with the use of a vertically homogeneous cloud structure in the retrieval.

However, we highlight that a number of assumptions are used in these derivations. The $f_{ad}$ in Eq. (1) is assumed to be constant with height, but another important factor is the ratio:

$$k = \frac{r_v^3}{r_e^3} \qquad (3)$$

This relates the volume mean equivalent radius $r_v$ to the effective radius relevant for radiative transfer calculations. It is related to the width of the droplet size distribution and in-situ observations place $k$ around 0.80 (Martin et al., 1994; Pawlowska and Brenguier, 2000) in marine clouds. Although it has also been observed to vary with height (Painemal and Zuidema, 2011), our derivations assume it to be constant.

Relationships derived from the subadiabatic cloud model can be used to retrieve $N_d$ and $H$ from MODIS $\tau$ and effective droplet radius ($r_{e,M}$). Simulations by Platnick (2000) suggest that the retrieved $r_{e,M}$ is smaller than $r_{e,ad,t}$ as the channel weighting functions are below cloud top, but that the ratio depends on the MODIS channel used, $r_e$ profile and somewhat on the cloud optical depth. If the MODIS retrieval performs similarly to those simulations, then $r_{e,M}$ is similar to $r_{e,h}$ according to the results in Platnick (2000) Table 3a. The subadiabatic model also provides a simple relationship between $LWP$ and $H$:

$$LWP = \frac{1}{2} f_{ad} c_w H^2 \qquad (4)$$

allowing any $LWP$ product to be converted into a subadiabatic cloud thickness. However, aircraft measurements show a wide range of values for the assumed parameters, such as those related to mixing (Wood, 2005), which can bias MODIS retrievals (Painemal and Zuidema, 2011).

A term-by-term error analysis estimated $H$ could be estimated from space to within ±20 % (Bennartz, 2007), but validation is challenging due to the difficulty of retrieving $H$ of these clouds. Active instruments can profile many cloud types, and $H$ has been obtained from surface-based lidar and radar measurements as part of the U.S. Department of Energy Atmospheric Radiation Measurement (ARM) programme, as first demonstrated in Dong et al. (1997). Ceilometers alone allow high precision determination of cloud base (Dong et al., 2002), which could be combined with CALIPSO cloud top to provide $H$. A long-term oceanic ARM site is located on the Azores (the Eastern North Atlantic, ENA site) and a site was also on Ascension Island from June 2016—October 2017.

Greater geographic coverage has been obtained over the oceans through ship-based measurements such as the Marine ARM GPCI Investigation of Clouds (MAGIC) experiment, which included measurements taken from ships between Los Angeles, California and Hawaii. In principle, $H$ can be also be obtained from cloud radar mounted on aircraft that are commonly used in airborne campaigns (Wood et al., 2011; Zuidema et al., 2016).

Unfortunately, these instruments are not widely deployed and there is no consistent large-scale, continuous record of marine cloud thicknesses available from surface or airborne measurements. We cannot use the Azores or Ascension island ARM datasets for validation because no OCO2CLD-LIDAR-AUX retrieval occurs within ±0.5° (70—80 km) of their locations, and we cannot use MAGIC retrievals of clouds base in concert with CALIPSO since the ship measurements finished before the launch of OCO-2. Future coverage may be improved through the development of more compact and efficient ceilometers

that could be widely attached to buoys as in Mariage et al. (2017), or on seafaring autonomous vehicles (Meinig et al., 2015).

Spaceborne sensors offer unsurpassed coverage of ocean clouds but current spaceborne capabilities to retrieve $H$ of marine boundary layer clouds are limited since these clouds are often optically thick enough to attenuate lidar, and with $H$ from

$10^2$—$10^3$ m, radars need higher vertical sampling than that offered by, for example, CloudSat's downsampled 240 m bins. Lidar can, however, be used to estimate the cloud top droplet number density for clouds with narrow droplet size distributions. This has been demonstrated using retrievals based on combined measurements from CALIPSO and the Polarization and Directionality of the Earth Reflectance (POLDER) instrument (Zeng et al., 2014). This can be related to $H$ when combined with the same assumptions used in MODIS retrievals.

**2.2 Explicit retrieval of cloud thickness using photon path length**

An alternative approach to retrieve cloud thickness is to consider measurements which are sensitive to photon path length through differential absorption between channels. The basic principle of these retrievals is that if an instrument measures channels with similar wavelengths but different molecular absorption coefficients, then in the absence of atmospheric emission or scattering into the beam their radiances will both be described by Beer's Law:

$I = I_0 \exp(-k\Delta z)$                                                        (5)

Where $I$ is the measured radiance, $I_0$ the initial radiance, $k$ the extinction coefficient and $\Delta z$ the photon path length. In the case of reflectance measurements such as those from OCO-2, the narrow wavelength range tends to mean that $I_0$ and $\Delta z$ are similar between channels, and therefore from measurements of $I$ and spectroscopic information for $k$, the photon path length can be derived. In a uniform cloud scene of sufficient optical depth, the photon path will consist of an above-cloud path that

changes with $P_{top}$, and a within-cloud path that depends on the within-cloud scattering, which is in turn related to $H$ and $\tau$. The use of photon path length information for cloud top pressure retrievals was suggested as early as the 1960s (Hanel, 1961; Yamamoto and Wark, 1961), and in more recent decades the possibility of probing within-cloud structure has been developed (Li and Min, 2010; Min et al., 2004).

The oxygen A-band on OCO-2 is a good candidate for this sort of measurement, as it relies on absorption by a ubiquitous

and well mixed atmospheric constituent whose fractional abundance does not greatly vary in space or time. From a single measurement using two channels, only a single piece of information on photon path can be obtained, and this is related to the total photon path. However, by combining multiple angles, channels or bands it has been proposed that obtaining both $P_{top}$ and $H$ is possible. Suggestions have included multi-angle measurements such as in Merlin et al. (2016) or both the A- and B-band as in Yang et al. (2013).

For the purpose of marine stratocumulus, however, neither of these approaches have proven to be tenable. The multiangular results in Merlin et al. (2016) refer to clouds with $H > 2$—3 km and an updated information content analysis accounting for the on-orbit performance of the A- and B-Band sensors of the Earth Polychromatic Imaging Camera (EPIC) on the Deep

Space Climate ObserVatoRy (DSCOVR) has concluded that "only cloud top height can be reliably inferred" (Davis et al., 2018).

There is also a heritage of considering high spectral resolution measurements in the oxygen A-band to obtain $H$. Early work considered cloud top height (Fischer and Grassl, 1991; Yamamoto and Wark, 1961) and later estimated the required spectral resolution required to allow separation of the above- and within-cloud components to allow retrievals of both $P_{top}$ and $H$ (Heidinger and Stephens, 2000; O'Brien and Mitchell, 1992; Stephens and Heidinger, 2000). These suggested that a spectral sampling of 0.5—1.0 cm$^{-1}$ is necessary for a joint retrieval, similar to the 0.5 cm$^{-1}$ that Min and Harrison (2004) estimated as necessary to obtain four pieces of information in an atmosphere with optically thin scattering layers. These results are dependent somewhat on other instrument characteristics such as the signal-to-noise ratio. Channels with stronger absorption tend to be more sensitive to $P_{top}$ and those with weaker absorption to $H$, since the more strongly absorbing channels tend to see complete extinction if their photons are multiply scattered within the cloud. Higher spectral resolution means channels with thinner spectral width that cover a smaller range of absorption coefficients, and this improves sampling in terms of oxygen absorption coefficient so aids in distinguishing between $P_{top}$ and $H$ contributions.

However, spectral resolution is not the only limiting factor: instrumental noise and other uncertainties such as those associated with the vertical profile of atmospheric moisture above the cloud, can also limit a retrieval. Information content analyses have shown theoretical differences in performance between the Global Ozone Monitoring Experiment-2 (GOME-2, Munro et al. (2016)) and OCO-2 instruments. GOME-2 has spectral sampling of approximately $\Delta\lambda$=0.21 nm and a full width at half maximum (FWHM) near 0.50 nm, compared with OCO-2's $\Delta\lambda \approx 0.02$ nm and FWHM $\approx 0.04$ nm (approximate values, they vary with wavelength due to instrumental design). GOME-2's typical Signal to Noise Ratio (SNR) is near 100, whereas OCO-2's continuum channel SNR typically ranges from 400—800 in the A-band, although it is larger in absorption bands.

Considering only instrumental SNR, it was found that GOME-2 is not able to retrieve $H$ in addition to $P_{top}$ (Schuessler et al., 2014), however OCO-2 is able to retrieve both for horizontally homogeneous clouds even after accounting for error covariance terms due to uncertainty in $r_{e,h}$ and the temperature and humidity profiles (Richardson and Stephens, 2018). This analysis determined that a micro-window of 75 OCO-2 channels contained sufficient information for a three-property joint cloud retrieval.

Example simulated cloudy scene spectra for GOME-2 and OCO-2 are shown in Figure 1 and the 75 channels used in OCO2CLD-LIDAR-AUX are highlighted in red. Also displayed are responses for each instrument and cloud property, sorted by the channel-mean oxygen absorption coefficient in order to emphasise how spectral response depends on this.

The cloud optical depth Jacobian is shown as $\partial I/\partial\tau$ where $I$ refers to each channel's modelled radiance. $H$ is expressed as $\Delta P_c$, in terms of atmospheric pressure coordinates in hPa. The $P_{top}$ and $\Delta P_c$ Jacobians are shown as the response in $I/I_c$, where $I_c$ is the continuum radiance.

Inspection of Figure 1(b—d) shows similar responses to optical depth across the two instruments, but a clear difference for the $P_{top}$ and $\Delta P_c$ responses. In particular, the GOME-2 Jacobians are more similar to each other than those of OCO-2. For

OCO-2 the greatest change in fractional absorption, represented by the deepest trough in the $I/I_c$ Jacobian, occurs in more strongly absorbing channels for $P_{top}$ compared with $\Delta P_c$, as described previously. It is this difference in Jacobians that results in independent information that allows retrievals of both $P_{top}$ and $H$ with OCO-2.

## 3 OCO-2 mission, instrumentation and orbit

We summarise relevant details of the OCO-2 orbit, viewing modes and instrumentation here, full descriptions are in the Level 2 Full Physics Algorithm Theoretical Basis Document (L2FP ATBD, Boesch et al. (2017)).

OCO-2 leads the A-train constellation (L'Ecuyer and Jiang, 2010) and over the OCO2CLD-LIDAR-AUX coverage from 2014-09-06 to 2016-12-30 it followed the CloudSat reference ground track (RTG) approximately 7 minutes ahead of CALIPSO and approximately 217 km to the east of the Aqua RTG in the ascending node. The A-train is in a Sun-

synchronous orbit with an ascending equatorial crossing time near 1:30 pm and an equatorial repeat time of approximately 16 days.

The OCO-2 operational science viewing modes are nadir and glint, with glint preferred for ocean XCO2 retrievals due to improved SNR. Nadir, however, offers the advantages of collocation with the near-nadir CloudSat and CALIPSO views, plus a shorter path through the atmosphere which allows more signal from absorbing channels.

Originally orbits were alternated between nadir and glint view before some ocean-dominated orbits were always committed to glint. OCO2CLD-LIDAR-AUX only uses nadir orbits where collocation with CALIOP is possible.

The satellite operates in an angled pushbroom fashion with an 8-footprint swath whose orientation rotates through the orbit to optimise solar panel output. Footprint geometry varies but at nadir is approximately 1.4 km×2.2 km. Each footprint is measured by three co-boresighted Fourier Transform grating spectrometers in the $O_2$ A-band, weak $CO_2$ band ($\lambda\sim1.61$ μm)

and strong $CO_2$ ($\lambda\sim2.06$ μm) band. Spectral sampling varies from 0.01—0.02 nm in wavelength and instrument line shape full width at half maximum (ILS FWHM) is approximately 0.04 nm. A-band Signal-to-Noise Ratio (SNR) typically ranges from 400—800 in continuum channels.

## 4 Retrieval design and data sources

OCO2CLD-LIDAR-AUX uses an iterative optimal estimation scheme in which a prior cloud state is updated such that the

simulated radiances associated with that state agree with the measured OCO-2 spectrum, given appropriate weighting to uncertainties in both the prior and observations.

### 4.1 Optimal estimation principles

For each footprint where a retrieval is attempted we construct a cloud state vector $\boldsymbol{x} = \left[\ln \tau \;\; \ln P_{top} \;\; \ln \Delta P_c\right]^T$ and an observation state vector $\boldsymbol{y} = [I_1 \;\; I_2 \ldots I_{75}]^T$ where each $I_i$ ($i = 1,\ldots,75$) is a measured channel radiance. $\Delta P_c$ is the cloud

geometric thickness $H$ expressed in terms of change in atmospheric pressure. We begin with a prior cloud state vector whose components are Gaussian, represented by a mean state vector $\boldsymbol{x_a}$ and covariance matrix $\mathbf{S_a}$. Meanwhile the observational uncertainty is represented by a zero-mean Gaussian with covariance matrix $\mathbf{S_\epsilon}$. Optimal estimation produces a maximised posterior probability density of the posterior state given both the prior state and the observations, with appropriate weighting

for their relative uncertainties. In our case the individual contributions to observational uncertainties are assumed to be independent and thus add in quadrature such that $\mathbf{S_\epsilon}$ is simply the sum of each term's covariance. The posterior is estimated by applying Bayes' theorem assuming a locally linear forward model encapsulated in the Jacobian matrix $\mathbf{K}$ whose elements are $K_{i,j} = \partial y_i / \partial x_j$. The solutions for the posterior state and its covariance in a totally linear case are (Rodgers, 2000):

$$\hat{\boldsymbol{x}} = \boldsymbol{x_a} + \mathbf{S_a}\mathbf{K}^T(\mathbf{K}\mathbf{S_a}\mathbf{K}^T + \mathbf{S_\epsilon})^{-1}(\boldsymbol{y} - \mathbf{K}\boldsymbol{x_a}) \tag{6}$$

$$\hat{\mathbf{S}} = (\mathbf{K}^T\mathbf{S_\epsilon}^{-1}\mathbf{K} + \mathbf{S_a}^{-1})^{-1} \tag{7}$$

Nonlinearity is addressed by allowing multiple iterations, where step $n+1$ properties are related to the prior step $n$ via:

$$\boldsymbol{x}_{n+1} = \boldsymbol{x_a} + \mathbf{S_a}\mathbf{K}_i^T\big(\mathbf{K}_i\mathbf{S_a}\mathbf{K}_i^T + \mathbf{S_\epsilon}\big)^{-1}[\boldsymbol{y} - F(\boldsymbol{x}_i) + \mathbf{K}_i(\boldsymbol{x}_n - \boldsymbol{x_a})] \tag{8}$$

where $F$ is the forward model. Table 1 lists the sources for each element of these matrices: prior $\tau$ uses a lookup table based on A-band radiances, the $P_{top}$ is derived from CALIPSO and $\Delta P_c$ from Equation (1) with the prior $\tau$ and assumed $r_{e,h} = 12$

$\mu$m.

The retrieval takes the first step equal to the prior, i.e. $x_0 = x_a$ and then iterates up to six times using the posterior of the previous step as the starting point of the next iteration. The step with the lowest $\chi^2$ is reported as the retrieved state, and the number of iterations was selected to balance quality of convergence and computational expense. No explicit convergence criterion was adopted: all retrievals that did not trigger computational problems are reported and users are provided with both

the state estimate and the $\chi^2$ for the retrieval step used. Synthetic retrieval tests showed typical convergence with 2 steps (Richardson and Stephens, 2018), and 58 % of successful retrievals used step 1 or 2. Only around 10 % of retrievals selected step 6.

## 4.2 Forward model

We use the OCO-2 Level 2 Full Physics retrieval algorithm's radiative transfer model (henceforth L2RTM). This is a

multiply scattering line-by-line radiative transfer model based on VLIDORT (Spurr, 2006) with a modified 2 orders of scattering code (2OS, Natraj and Spurr (2007)) and is available from GitHub (https://github.com/nasa/RtRetrievalFramework ). It was designed for clear skies but has previously been modified for cloudy scene simulations (Richardson et al., 2017). Clouds are treated as homogeneous plane-parallel layers and scattering properties are from Mie scattering calculations using a gamma distribution with characteristic $r_e = r_{e,h} = 12$ $\mu$m (see Section 5.3.3 for discussion on sensitivity to this choice). This

selection is based on the median MODIS droplet size retrieved using the 2.1 $\mu$m channel for our selected cloud cases (Nakajima and King, 1990). The median MODIS $r_{e,M}$ is 12.5 $\mu$m with a 14—86 % range of 8.4—18.6 $\mu$m, we use the closest integer value since pre-calculated Mie tables are available only for integer $r_e$ values in the L2FP code.

It has been shown that for typical cloud retrieval channels, vertically homogeneous clouds have similar radiative properties to subadiabatic clouds of the same $H$, $LWP$ and $\tau$ provided that their $r_{e,h}$ is 5/6ths that of the subadiabatic cloud top $r_{e,ad,t}$ (Brenguier et al., 2000). As discussed in Section 2.1 the retrieved $r_{e,M}$ is sensitive to within-cloud rather than cloud-top $r_e$ (Platnick, 2000), except where non-adiabatic influences such as entrainment of dry air and evaporation reduces cloud-top $r_e$

(Nakajima et al., 2010). Since the appropriate $r_{e,h}$ is 5/6ths of $r_{e,ad,t}$, and these MODIS retrievals sample within the cloud where droplets are smaller, we expect $r_{e,M}$ to approach $r_{e,h}$. Regardless, we test the sensitivity of the retrieval to $r_{e,h}$ in Section 5.3.3.

A fixed $r_{e,h}$ was selected to speed computation: the L2RTM requires a full extinction profile for every sounding and every $r_{e,h}$ used for any sounding in the orbit. Limiting this to one option speeds the retrieval and its effect on the retrieval is included

by adding a term to the observation covariance (Richardson and Stephens, 2018).

Pressure levels are assigned to cloud top, centre and bottom and 17 other layers are linearly interpolated to the top of atmosphere or surface. An extinction coefficient is assigned to the cloud centre and interpolated uniformly between the cloud top and bottom. Jacobians are calculated numerically using finite differences as follows: (1) for $\tau$, the extinction is scaled, (2) for $P_{top}$ all three cloud pressure levels have their pressure increased by $\delta P$, (3) for $\Delta P_c$ the bottom level pressure is

increased by $\delta P$, the centre level by $0.5\delta P$ and the extinction is scaled to maintain constant $\tau$. The size of each increment is in Table 1.

The L2RTM also requires information about location, geometry, meteorology, satellite orbital parameters and instrument characteristics. For this we use the OCO-2 files associated with Version 7 of the XCO2 product, since OCO2CLD-LIDAR-AUX processing began before the version 8 release. We use Level 1b science spectra (L1bSc) and interpolated weather

forecasts from the European Centre for Medium-range Weather Forecasts (ECMWF) which provides temperature and humidity profiles along with surface wind speed for the Cox-Munk sea surface reflectance model (Cox and Munk, 1954). For $O_2$ absorption we use the OCO-2 version 5 absorption coefficient (ABSCO) tables (Drouin et al., 2016) which were used in version 8 of the OCO-2 XCO2 retrieval. They better represent oxygen absorption in several ways compared with those used in version 7, including through the handling of line mixing and collision induced absorption. Input properties used,

version numbers and key citations are provided in Table 2.

### 4.3 Algorithm design

The algorithm first attempts to identify liquid clouds over the ocean, then assigns a prior and iterates to a posterior state using Equation (6). OCO-2's original cloud screening algorithm was not designed for nadir view over the oceans. Given that non-cloudy scenes are dark in nadir view, we collect A-band and weak $CO_2$ continuum radiances $I_c$ from the L1bSc spectra

and divide by $\mu_0 = \cos(SZA)$. Then the 10 channels that were, on average, brightest over November 2015 are taken as the continuum with a separate set for each footprint across the swath. If $\mu_0^{-1}I$ exceeds a threshold in each channel (A-band $\mu_0 I_{O2} = 6\times10^{19}$, weak $CO_2$ band $\mu_0 I_{wk}$ $1\times10^{19}$ photons m$^{-2}$ s$^{-1}$ sr$^{-1}$ μm$^{-1}$) then a cloud is flagged and agreement with the MODIS

"confident" cloud flag is approximately 85 %. This threshold is equivalent to just over 15 W m$^{-2}$ sr$^{-1}$ $\mu$m$^{-1}$, compared with the median $\mu_0^{-1}I$ near 4 W m$^{-2}$ sr$^{-1}$ $\mu$m$^{-1}$ in clear sky conditions, according to the OCO-2 A-band preprocessor.

A further constraint is provided by CALIPSO: the nearest CALIPSO footprint is checked, and the footprint is only used if CALIPSO identifies a single-layer cloud whose $P_{top}$ > 680 hPa. The CALIPSO $P_{top}$ threshold limits our sample to the low
cloud threshold of the International Satellite Cloud Climatology Project (Rossow and Schiffer, 1991) and helps to filter out non-liquid clouds. CALIPSO also helps to exclude some multi-layer cloud cases which violate our retrieval assumption of a single layer cloud.

If both of these tests agree on a cloud, then the continuum A-band and weak CO$_2$ $\mu_0I$ are used to estimate cloud phase via a lookup table that exploits how ice absorbs more strongly than water in the weak CO$_2$ band relative to the A-band (Nakajima
and King, 1990). A lookup table is also used to estimate the prior cloud optical depth from the continuum A-band radiance, since more optically thick clouds tend to be brighter, and Figure 2 shows both the phase and $\tau$ lookup tables. Other prior properties and covariance terms are assigned as described in Table 1. For observation covariance, footprint SNR is added to pre-calculated matrices that are scaled for $\mu_0$ and cloud $\tau$ as described in Richardson & Stephens (2018). These matrices include terms related to uncertainty in atmospheric temperature and moisture, plus cloud $r_{e,h}$. **Error! Reference source not**
**found.**Figure 2 is a flow diagram that summarises the full retrieval.

## 4.4 Product output fields and collocation with CloudSat

OCO2CLD-LIDAR-AUX is both a demonstration of hyperspectral A-band cloud retrievals and an attempt to fill in a gap in information about marine boundary layer clouds. It aims to allow this new information to be easily linked to CloudSat, CALIPSO and other mission data for cloud process studies.
The output files are therefore reported following the standard CloudSat granule structure as part of the Release 5 (R05) data products available from the CloudSat Data Processing Center (http://www.cira.cloudsat.colostate.edu) along with an Interface Control Document that details all of the data structures.

The retrieved cloud properties $\tau$, $P_{top}$ and $\Delta P_c$ are collocated with the CloudSat footprints by minimising surface distance between the centre of each instrument's footprint according to a nearest neighbour great circle scheme.. Collocation
introduces some inconsistencies which we refer to as "collocation error". For OCO-2 and CloudSat, difference in overpass times and footprint shape & size are the main factors, but for validation against MODIS additional contributions may occur due to the different viewing geometries causing parallax error.

Cloud retrievals are provided for a CloudSat footprint whenever the distance between the footprint centres is < 1.25 km, or approximately half the length of an OCO-2 or CloudSat footprint. Since OCO-2 has a swath of 8 footprints and CloudSat
does not, most of the OCO-2 data are not matched to CloudSat. Therefore the native OCO-2 footprint structure is also reported, and these are distinguished by the inclusion of full_swath in the dataset name. For example, the dataset Cloud_Optical_Depth is the OCO-2 estimate of $\tau$ collocated with CloudSat, whereas full_swath_Cloud_Optical_Depth is the

OCO-2 includes all OCO-2 footprints. Furthermore, contextual information as solar zenith angle and the local variance of the A-band continuum radiance, along with collocation indexing and matchup distances are provided. This allows users to include more swath information or apply their selected matchup criteria.

Finally, a Quality_flag is provided whose components are described in Table 3. When a retrieval is attempted the Quality_flag is initialised to zero, and integer values are added as potential warning factors are identified such that Quality_flag = 0 represent the best quality data. The final value is the sum of all flags associated with the retrieval, and valid retrievals have a flag range of 0—7. It is recommended that any cases with Quality_flag $\geq$ 4 are also excluded, as this includes the very small fraction of soundings where the detector experienced a cosmic ray strike, resulting in non-physical spectral signatures. The retrieval statistics are split by Quality_flag and analysed in Section 5, which aims to provide the evidence that users need to decide which range of Quality_flag values they can accept.

## 4.5 Algorithm throughput and performance statistics

The average number of attempted retrievals with a < 10 km matchup distance to CALIPSO is 7,336 per orbit (~11 % of OCO-2 footprints). Table 4 shows statistics of the retrieval throughput and some comparisons to MODIS, which we use later for validation.

Retrieval failure occurs in 2.0 % of cases and includes incomplete or inconsistent input data and failures in the radiative transfer or optimal estimation retrieval code. 0.1 % are rejected due to cosmic ray strikes detected in our channels, and 4.6 % are outside the retrieval space, mostly a posterior state with a cloud bottom below Earth's surface. Also shown in Table 4 is that 84.5 % of OCO2CLD-LIDAR-AUX successful retrievals are identified as liquid clouds by MODIS, and 71.0 % have collocated MODIS optical depth retrievals. This subset is used for validation in Section 5.

The retrieval is computationally intensive and the cloud retrieval is not integrated into the OCO-2 L2FP clear sky XCO2 retrieval. Major bottlenecks include (1) reading and writing data necessary to link the L2FP radiative transfer code to the cloud retrieval and (2) the optimal estimation retrieval that includes inverting multiple 75×75 element matrices. The average orbit processing time is approximately one hour, and this is helped by the sub-selection of 75 channels from the 853 undamaged A-band channels that reduces the theoretical computational burden of the matrix calculations by around 99 %. Using the full spectrum in the retrieval results in an average computation time of >30 hours per orbit.

## 5 Cloud properties from OCO2CLD-LIDAR-AUX compared with MODIS and CALIPSO

## 5.1 Optical depth compared with MODIS

Cloud retrievals are best-suited for single layer, horizontally homogeneous clouds, with a low solar zenith angle. OCO2CLD-LIDAR-AUX's Quality_flag is designed to help users identify these cases. Firstly, the solar zenith angle can bias MODIS-retrieved $\tau$ for reasons including angle-dependent differences in real world radiative transfer including 3-D cloud effects, and the plane-parallel assumptions commonly used in retrievals. These and others are discussed in the

literature (Chambers et al., 2001; Grosvenor and Wood, 2014; Liang et al., 2015; Várnai and Marshak, 2002), and contribute to larger expected uncertainty in retrieved $\tau$ at higher solar zenith angles. The OCO2CLD-LIDAR-AUX Quality_flag increases the value of Quality_flag by 1 when the solar zenith angle, SZA > 45°.

Secondly, the ratio of the 10-channel continuum radiances in each band, $I_{wk}/I_{O2}$ is related both to cloud droplet size and cloud phase. Particularly when ice is present, there is increased absorption of weak $CO_2$ band radiance and therefore a lower ratio. While OCO2CLD-LIDAR-AUX estimates phase using the lookup table from Figure 2, this was optimised based on agreement with MODIS for a subset of orbits. We find that this is not strict enough, and approximately 10 % of the data with $I_{wk}/I_{O2} < 0.28$ show much greater discrepancies compared with MODIS and CALIPSO. This is likely because the cloud is not entirely liquid or due to the presence of overlying cirrus or aerosol layers, and therefore a retrieval that attempts to fit a liquid cloud model to the observed spectrum will result in biased cloud properties. For these cases we increase the Quality_flag value by 2.

We compare the statistics of OCO-2 minus MODIS $\tau$ retrievals in Figure 4 for the full sample (a,b), and subsets split according to their SZA and radiance ratio (c,d). Only footprints with valid OCO-2 and MODIS retrievals are included. Statistics are presented both for the absolute difference, $\delta\tau = \tau_{OCO-2} - \tau_{MODIS}$, and scaled by the MODIS reported optical depth uncertainty, $\delta\tau/\sigma_{\tau,MODIS}$. The distributions are non-Gaussian (full sample Kolmogorov-Smirnov test statistic = 0.279, $N > 1.6 \times 10^7$, p < 0.001) and skewed, so we report the median and the 14—86[th] percentiles in place of the mean and standard deviation. For the full sample in Figure 4(b) the median bias is 0.02 times the MODIS uncertainty and the 14—86 % range is -1.12 to 1.02. If collocation error were zero and errors between OCO-2 and MODIS were equal and independent, adding them in quadrature results in an expected range of -1.41 to 1.41. The smaller differences we find indicate that MODIS and OCO-2 errors may be correlated, and that the OCO-2 and/or MODIS uncertainties may be smaller than reported in the MODIS product.

For Quality_flag = 0, where SZA < 45° and $I_{wk}/I_{O2} < 0.28$, Figure 4(d) shows that the median bias relative to MODIS is 4 % of the MODIS reported error, which is slightly larger than the full-sample bias. However, the 14—86 % range is narrower at -87 % to +83 %. The OCO-2 derived optical depths are consistent with those from MODIS.

## 5.2 Cloud top pressure versus CALIPSO, geometric thickness versus MODIS and implied subadiabaticity

Next we investigate the retrieved $P_{top}$ and $\Delta P_c$ in Figure 5. The retrieved values are compared with their priors, which for $P_{top}$ means a comparison with CALIPSO and for $\Delta P_c$ a comparison with an adiabatic estimate based on the cloud's optical depth and an assumed 12 micron droplet size. As in the MODIS comparison, statistics are shown for the full dataset and for when it is subset by solar zenith angle and the $I_{wk}/I_{O2}$ radiance ratio.

The full sample distribution, particularly of $P_{top}$, is skewed and its apparently small median bias of 4 hPa is large relative to the assumed 5 hPa prior error in $P_{top}$. The median change in cloud pressure thickness is close to 0 hPa, so retrieved average cloud thickness matches the typical adiabatic cloud thickness.

When selecting Quality_flag = 0, the $\Delta P_c$ retrieval differences become more symmetric with an 18—86 % range of [-5,4] hPa. However, the retrieved $P_{top}$ bias is opposite to that of the full sample and shows that the OCO-2 retrievals result in increased cloud altitude relative to that seen by the precise CALIOP lidar. Our use of a tight prior constraint does ensure that that the discrepancy is smaller than for MODIS minus CALIOP, of -22 [-115, 57] hPa, but this nevertheless needs investigation.

Comparison of $\Delta P_c$ with an adiabatic prior is somewhat unsatisfying, given that we have no direct validation data to determine whether the OCO2CLD-LIDAR-AUX retrievals add extra value. Therefore we take an indirect approach based on comparison with MODIS cloud thickness. For this part we turn to geometric thickness $H$ in metres, converting our $\Delta P_c$ to $H$ using a standard scale height calculation (e.g. Equation 6 in Wood and Bretherton (2006)) and then conver MODIS retrieved LWP to $H$ using Equation (4) with the local condensation coefficient calculated for the OCO-2 cloud base height using the collocated ECMWF meteorology and taking MODIS $f_{ad}$ = 1, since this is currently not retrieved. Here we temporarily switch to using $H$ rather than $\Delta P_c$ because (i) the $\Delta P_c$ conversion would include using MODIS-derived $P_{top}$, which could introduce greater spread and (ii) we can relate $H$ directly to the cloud adiabatic fraction $f_{ad}$ for more insight into our retrieval's behaviour.

Following this, we have independent estimates for all three properties of our state vector, and 2d histograms of these comparisons are shown in Figure 6. Focussing on Figure 6(c), the OCO-2 based $H$ tends to be larger than that from MODIS, implying a less-adiabatic (i.e. lower $f_{ad}$) cloud.

One process that can reduce $f_{ad}$ is entrainment of dry air at the cloud top, which effectively dilutes the cloud and thereby increases its thickness. This entrainment is stronger when the overlying inversion is weaker, so to explore this we estimate $f_{ad}$ implied by our retrievals and bin this by estimated inversion strength (EIS, Wood and Bretherton (2006)). We apply Eq. (4), assuming that the OCO-2 and MODIS $LWP$ are equal, then rearrange to obtain:

$$f_{ad} \approx \sqrt{\frac{H_{OCO-2}}{H_{MODIS}}} \tag{9}$$

The retrievals are split by $\tau$ and then the median $f_{ad}$ in each EIS bin is calculated. EIS bins are: < 0 °C, then in 2 °C increments up to 16 °C, and finally >16 °C. The results in Figure 7(a) show a general tendency that OCO-2 implies increasingle subadiabatic conditions for weaker inversions, and the change from the prior to the posterior is shown in Figure 7(b).

For optically thinner clouds the posterior state is consistent with physical expectations, and the retrieval makes substantial changes from the prior. Meanwhile, it does not make substantial changes for optically thicker clouds, although the prior itself shows the expected $f_{ad}(EIS)$ relationship. This is due to differences between $r_{e,M}$ and the assumed $r_{e,h}$ = 12 $\mu$m used to derive our prior thickness, where MODIS $r_{e,M}$ correlates with EIS.

The EIS-dependence of inferred $f_{ad}$ in Figure 7 we take as evidence that our retrieval is updating our retrieved cloud thicknesses in a physically consistent way, and is providing information beyond that obtained from an adiabatic prior. While we argue that this suggests additional value from our retrieval, there remain substantial uncertainties and potential biases,

due to the lack of available true validation data as described in Section 2.1. Our first concern is the $P_{top}$ bias, as both the $P_{top}$ and $\Delta P_c$ retrievals respond to photon path length, and as such a bias in one may result in biases in the other.

## 5.3 Investigation of $P_{top}$ bias

The bias in $P_{top}$ is concerning given the tight constraint provided by CALIPSO. Identifying likely causes of the bias is a priority for interpreting the data and for improving future retrievals. As changes in both $P_{top}$ and $\Delta P_c$ change photon path length, it is also possible that biases in $P_{top}$ could cause counteracting biases in $\Delta P_c$. Factors that could influence the photon path length include (1) aerosol layers within the field of view, (2) a nonuniform cloud field and (3) differences in the extinction profile between the real clouds and the vertically homogeneous plane parallel clouds used in the radiative transfer. We investigate each of these possibilities here. Further factors that we do not investigate include (i) errors in spectroscopy that affect simulated photon path lengths, (ii) improper instrument calibration or unaccounted for drifts in this calibration. We note that OCO2CLD-LIDAR-AUX used the most up-to-date ABSCO tables that were available at the time of processing and that future product versions (including an under development OCO-2 only retrieval) will use the latest calibrated spectra available.

### 5.3.1 Aerosol layers

We use the collocated CALIPSO 05kmALay product to identify potential aerosol layers in the OCO-2 field of view (Omar et al., 2009; Vaughan et al., 2009). This product uses multiple lidar shots averaged along track to help identify optically thin layers that would not be reliably detected by a single shot. This product provides estimates of layer optical depth at 532 nm and 1064 nm as well as layer location. Here the results are separated according to whether the aerosol layer is detected above the 01kmCLay cloud or below it. Detection below the cloud is possible when there is a broken cloud scene, such that the 1 km product returns a cloud within the OCO-2 field of view but the 5 km product detects aerosol elsewhere.

Aerosol is ubiquitous in Earth's atmosphere and particularly thin layers are not detected by CALIPSO, even with extensive averaging. However, the OCO-2 radiances should only be weakly affected if a layer is thin enough to avoid detection by CALIOP.

The results are in Figure 8~~Figure 6~~, and are split by cloud optical depth above or below 5. While aerosol layers above the cloud do result in larger biases, the median shifts by 2.2 hPa for optically thin clouds and 0.6 hPa in optically thick clouds, both of which are less than half of the total $P_{top}$ bias. For optically thin clouds, above cloud aerosol notably increases the number of strongly biased cases, shifting the 14[th] percentile from -20.8 hPa to -42.8 hPa. This shift fits with aerosol layers above the cloud shortening photon path lengths, and is inconsistent with a dominant role for increased surface reflection in scenes with a low value of retrieved cloud $\tau$, since surface reflections would increase photon path length and therefore retrieved $P_{top}$. Overall, we conclude that the presence of aerosol only has a small effect on the median retrieval and there is no evidence that it is the dominant cause of the $P_{top}$ bias relative to CALIPSO.

### 5.3.2 Horizontal spatial variability

To quantify cloud spatial variability we take the standard deviation of the A-band continuum radiance in all neighbouring OCO-2 footprints and divide this by the retrieval footprint's continuum mean radiance as described in Richardson et al. (2017). This is not a direct estimate of the within-footprint variability, but is likely positively correlated with it. Furthermore, footprint-to-footprint variability will also indicate where larger collocation errors are likely. Cloud motion between the OCO-2 and CALIPSO overpass times, or georeferencing errors will also be larger when this parameter is higher.

In Figure 7 we show evaluations of the retrieval ($\tau$, $P_{top}$, $\Delta P_c$) properties split by deciles in the radiance spatial variability. Only results in which the Quality_flag = 0 retrieval are shown. For decile 1, the most horizontally homogeneous cloud fields, posterior $\tau$ and $\Delta P_c$ are both reduced relative to the prior. These have opposing effects on the within-cloud photon path length. The median reduction in the extinction coefficient used in the lowest decile of retrieved clouds is 1 %, implying a minor change in within-cloud path between the prior and posterior states. However, they experience the largest changes in $P_{top}$, implying that the spectra support shorter mean photon path lengths than those implied by the prior, and that the retrieval is adjusting $P_{top}$ rather than within-cloud path length to match the spectra. Given the precision and reliability of CALIPSO, this points to another biasing factor.

Two conclusions can be reached from this result: 1) it is unlikely that cloud heterogeneity contributes to the overall positive bias in the retrievals and 2) there are most likely compensating errors related to 3 dimensional radiative transfer causing a reduction in bias as heterogeneity increases. The results so far imply that the OCO-2 spectra are consistent with a reduced photon path length relative to the a-priori cloud state used in the radiative transfer, and that the OCO2CLD-LIDAR-AUX retrieval is accounting for this by shifting the cloud tops upward. The collocated MODIS properties shown in Figure 9Figure 7(d,e) also change with the spatial homogeneity parameters, with a larger $\tau$ and $r_{e,h}$ in the OCO-2 retrievals than reported by MODIS, which would be the case if retrieval biases were being driven by $r_{e,h}$. Next we investigate whether changes in the $r_{e,h}$ or prior $\Delta P_c$ used in the radiative transfer affect the posterior $P_{top}$.

### 5.3.3 Assumed cloud structure, prior $\Delta P_c$ and $r_{e,h}$

To test the effect of the assumed $r_{e,h}$ and prior $\Delta P_c$ we select 10 orbits which had a large number of Quality_flag = 0 footprints and re-run the retrieval four times: once each with $r_{e,h} = 6$ $\mu$m, $r_{e,h} = 18$ $\mu$m, and once each with $\Delta P_{c,prior}$ scaled by 0.5 and 2.0. This results in $N = 64,572$ Quality_flag = 0 retrievals in each case, whose statistics are compared in Figure 10Figure 8. This subsample has a particularly large median $P_{top}$ difference relative to CALIPSO of -10 hPa and changes of ±50 % in $r_{e,h}$ have only a minor effect of ±1 hPa in the median retrieved $P_{top}$, consistent with the values from Richardson & Stephens (2018). However, the change in prior $\Delta P_c$ has a substantial effect, scaling by a factor of 0.5 almost eliminates the median bias, although at the expense of increased spread and the appearance of increased bimodality. Scaling by a factor of 2.0 increases the $P_{top}$ bias.

With regards to $\Delta P_c$, for very large prior values the retrieval generally attempts to reduce the posterior value. This is promising, as in the ×2.0 prior thickness case, the values are unrealistically large for an average case and the retrieval is bringing them closer to reality. Despite the reductions, the posterior cloud thicknesses are unrealistically large for the ×2.0 case and unrealistically small for the ×0.5 case. Below we discuss how this may mean the retrieval is obtaining an equivalent optical $\Delta P_c$ for a vertically homogeneous cloud when the observed scene is of a vertically-nonuniform cloud. In the default thickness case the 14—86 % range in posterior minus prior $\Delta P_c$ is [-5,1] hPa, i.e. the inclusion of OCO-2 spectral information results in a thinning of the retrieved cloud, as in the unrealistically large case. When the prior thickness is halved the difference between prior and posterior $\Delta P_c$ is visually far less skewed with a range [-2,3].

We propose that the sensitivity in retrieved properties to changing $r_{e,h}$ and prior $\Delta P_c$ is representative of their sensitivity in general terms to the scattering phase function and the vertical profile of extinction coefficient $\beta(z)$ within the cloud.

The $\beta(z)$ structure is more heavily weighted towards the top of the cloud in a subadiabatic cloud model, for a given $\tau$, $H$ and *LWP* (see Supplementary Section 2, Figure S1). This means that photons that enter a subadiabatic cloud will, on average, tend to travel shorter distances before exiting the top of the cloud than an equivalent homogenous cloud and therefore the associated spectrum should have brighter absorption band channels. We propose that there is therefore a vertically homogeneous cloud bias introduced into the retrieval that likely manifests through reductions in retrieved $P_{top}$, and that this likely plays a substantial role in the 4—5 hPa bias in retrieved $P_{top}$. Further modifications to the radiative transfer interface would be required to investigate this, so it is a target of future research.

This result differs from previous work such as Brenguier et al. (2000) because the derivations looked at typical retrieval channels where atmospheric absorption was negligible over within-cloud distances. Future work will investigate whether a scaled vertically homogeneous cloud model may be used to approximate a subadiabatic cloud, or whether the radiative transfer must directly include nonuniform extinction profiles.

## 6 Retrieval statistics and maps

In Figure 11Figure 9 the full histograms of the retrieved cloud properties are shown and split by Quality_flag. Generally, the highest quality data cover clouds with a median optical thickness near 6 and a 14—86 % range of 2—15, while poorer quality retrievals have higher optical depths. Cloud optical depth increases poleward of the subtropics, coinciding with higher SZA, and also with higher prevalence of mixed phase clouds, which may be misidentified and result in high equivalent liquid optical depths. The best quality data also tend to be associated with clouds that are lower in the atmosphere and with a medium pressure thickness of 25 hPa, (i.e. $H$ near 250 m).

The full period mean retrieved properties are mapped in Figure 12Figure 10 from 60 °S—60 °N on a 4°×4° latitude-longitude grid, along with the extinction coefficient $\beta_{(ext,h)} = \tau/\Delta P_c$. This is the $\beta_{ext,h}$ for the retrieved vertically homogeneous clouds and its interpretation is discussed below. This figure shows the lack of coverage in the central Pacific, where OCO-2 consistently measures in glint only mode to optimise for its XCO2 retrieval. Its current mode of operation

includes more ocean-dominated orbits committed to glint-only, but due to the importance of Alaska and Europe for the carbon cycle, nadir measurements of the stratocumulus decks off the coasts of Peru, Namibia, California and Australia continue.

The well known geographic structures of cloud properties are visible, with coastal stratocumulus and increases in optical
depth towards 60 °S/N. This is unsurprising given the use of the CALIPSO prior and the general agreement with MODIS. Changes relative to the prior state are mapped in Figure 13~~Figure 11~~. Figure 13~~Figure 11~~d shows the change in cloud pressure thickness from the prior, with expected patterns of relatively thinner clouds in the stratocumulus regimes, switching to thickening from the trade cumulus to cumulus regimes. For an adiabatic cloud model, $N_d \propto \tau/H^{-5}$, so these results indicate a tendency for higher droplet number densities in stratocumulus regions relative to convective regions, in agreement with
estimates from field measurements and theoretical expectations. This is reflected by how, in general, the $\beta_{ext,h}$ posterior is higher than the prior in the marine stratocumulus decks. The shift in $P_{top}$ relative to the CALIPSO prior, which we (Section 5.3.3) linked to $\beta_{ext}(z)$ differences between vertically homogeneous and adiabatic profiles, is larger and more negative in the stratocumulus regions too. This fits our proposal and may mean that changes in $\Delta P_c$ and $\beta_{ext}$ relative to the prior are similarly underestimated. Nevertheless, the stratocumulus regions are clearly visible and show changes consistent with our
theoretical understanding, so we propose that this is the first detection of internal cloud structure information from hyperspectral A-band retrievals.

**7 Discussion and Conclusions**

This paper introduced and described the OCO2CLD-LIDAR-AUX product in which CALIPSO's lidar provides a tight prior constraint on $P_{top}$ and an optimal estimation method then exploits OCO-2's hyperspectral A-band reflectance measurements
to attempt a retrieval of cloud geometric thickness in addition to $\tau$ and $P_{top}$. Output is provided to match the standards of the CloudSat Data Processing Center and the product will be downloadable there as part of the CloudSat data release 5. Collocation with CloudSat allows direct multi-sensor investigation of these cloudy scenes, but the product also provides the full OCO-2 swath data. OCO-2 switches between nadir and glint view orbits, and only those in which OCO-2 is in nadir are processed.

The product provides great potential to explore the thickness of marine stratocumulus clouds on a large scale and the consistency of those observations with commonly assumed cloud vertical structures used in passive solar reflectance cloud retrievals. This will greatly enhance the very limited targeted airborne campaigns whose sampling is limited. Other retrievals exist but with their own potential uncertainties: MODIS relies strongly on an assumed cloud structure while CALIPSO estimates are dominated by the cloud top. Hyperspectral A-band retrievals are based on photon path length and as such
provide a physically independent method of obtaining thickness information.

Retrieval of optical depth showed good agreement with MODIS, but cloud top pressure showed a small negative bias that was strongest in the most uniform clouds that tend to occur in the subadiabatic regions. This likely leads to biases in the

retrieved within-cloud path and therefore in inferred $H$, but through investigation of potential contributing factors we were able to rule out strong contributions from the constant assumed $r_{e,h}$ or the presence of aerosol layers detected by CALIPSO. Rather, we propose that the vertical extinction structure of the cloud is important, since the bias in $P_{top}$ scales with the prior $\Delta P_c$. Nevertheless, the small discrepancies relative to MODIS optical depth (Figure 4), the tendency to retrieve more

subadiabatic clouds under weaker inversions, at least for optically thinner clouds (Figure 7), and the increased extinction coefficient in the marine stratocumulus regions (Figure 13) suggest that the OCO-2 spectra add useful information. This information is obtained despite sub-selecting 75 out of the 853 functioning A-band channels based on a theoretical information content analysis. This reduces typical orbit processing time from >30 hours to 1 hour. A future version is under development using the OCO-2 A-band pre-processor code in which variable $r_{e,h}$ and integrated optical estimation and

forward modelling have been implemented. If successful, a future version of this data and/or an OCO-2 only retrieval will be provided. This OCO2CLD-LIDAR-AUX release uses Version 7 OCO-2 L1bSc spectra, whereas subsequent versions corrected some calibration errors associated with instrumental ice build up, which may introduce time-dependent instrumental errors that affect the retrieval.

We provide new information on marine stratocumulus clouds and identify a potential bias related to the cloud structure.

Future work will determine whether a bias correction is possible, and whether an equivalent vertically homogeneous extinction may be used to represent such clouds. While the retrieval is not directly sensitive to $r_{e,h}$, a retrieval of $r_{e,h}$ using OCO-2's $CO_2$ bands may be necessary in relating this equivalent vertically homogeneous structure to real-world cloud properties. This is a result of the within-cloud photon path information coming from molecular extinction, and therefore being excluded from the results reported in past studies of cloud structure and radiative transfer where the extinction was

dominated by clouds. Any other attempts to obtain within-cloud properties using extinction-derived within cloud paths should also consider the importance of the cloud's vertical extinction structure.

We conclude that there is evidence that OCO-2's cloudy scene footprints, that are screened and otherwise unused in the main OCO-2 products, contain useful cloud information for future investigations of marine boundary layer cloud properties. This potential is not limited to OCO-2, but includes operational or planned missions with similar A-band spectrometer

specifications such as GeoCarb (Moore et al., 2018).

**Supplement link:** see uploaded Supplementary Information PDF.

**Data availability:** The MOD06_L2 (Platnick et al., 2015) is available from the MODIS cloud product site at https://modis-

atmosphere.gsfc.nasa.gov/products/cloud and we use the version collocated with MODIS from Taylor et al. (2016), which also describes the CALIPSO data. The OCO2CLD-LIDAR-AUX dataset will be available from the CloudSat Data Processing Center at http://www.cloudsat.cira.colostate.edu/.

**Author contributions:** MR developed the overall retrieval code, performed the main analysis and wrote the paper. JL developed the optimal estimation code and helped with its integration. HQC provided collocation with CloudSat. JM provided the interface for cloudy radiative transfer simulations in the OCO-2 forward model. MDL assisted with product and analysis design. GLS proposed and directed the research. All authors contributed to planning and writing of the paper.

**Acknowledgements:** This research was carried out at the Jet Propulsion Laboratory, California Institute of Technology, under a contract with the National Aeronautics and Space Administration. Extra thanks go to Annmarie Eldering, Mike Gunson, David Crisp, Tommy Taylor and Chris O'Dell for helpful discussions, support and aid with OCO-2 data.

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

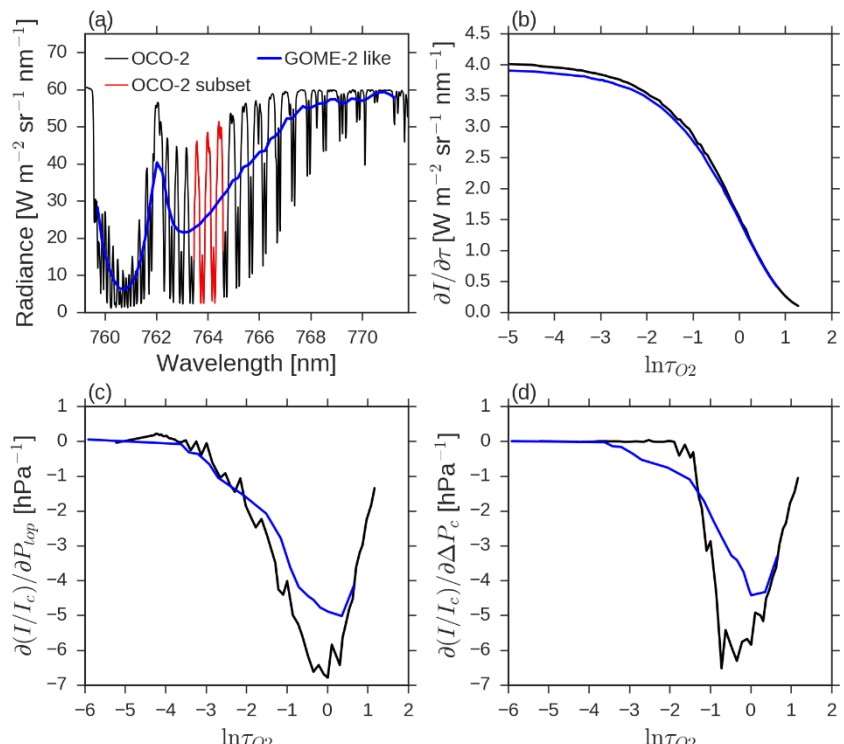

**Figure 1: (a) simulated A-band reflectance spectrum for a vertically homogeneous cloud with $\tau = 10$, $r_{e,h} = 12$ μm, $P_{top} = 750$ hPa as seen by OCO-2 and GOME-2-like sampling. The red OCO-2 subset refers to the 75 channels used in OCO2CLD-LIDAR-AUX. (b) OCO-2 and GOME-2 cloud $\tau$ Jacobians with channels organised by the baseline spectrum's channel mean molecular oxygen ln ($\tau_{O2}$). (c) $I/I_c$ Jacobians in response to $P_{top}$. (d) $I/I_c$ Jacobians in response to $\Delta P_c$.**

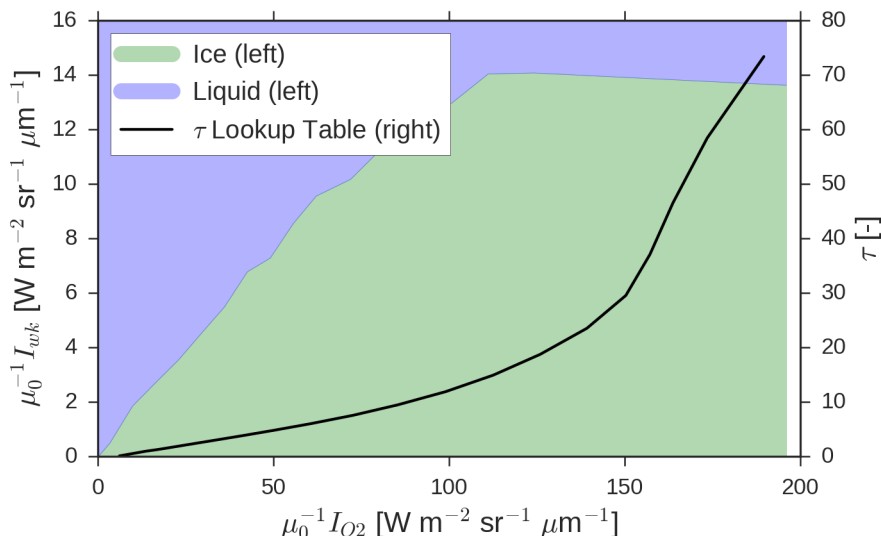

**Figure 2. Lookup tables for cloud phase and prior cloud $\tau$. Cloud phase is coloured for regions of SZA-corrected continuum A-band and weak CO$_2$ band radiances. Cloud $\tau$ is shown on the right axis and is only a function of the A-band radiances.**

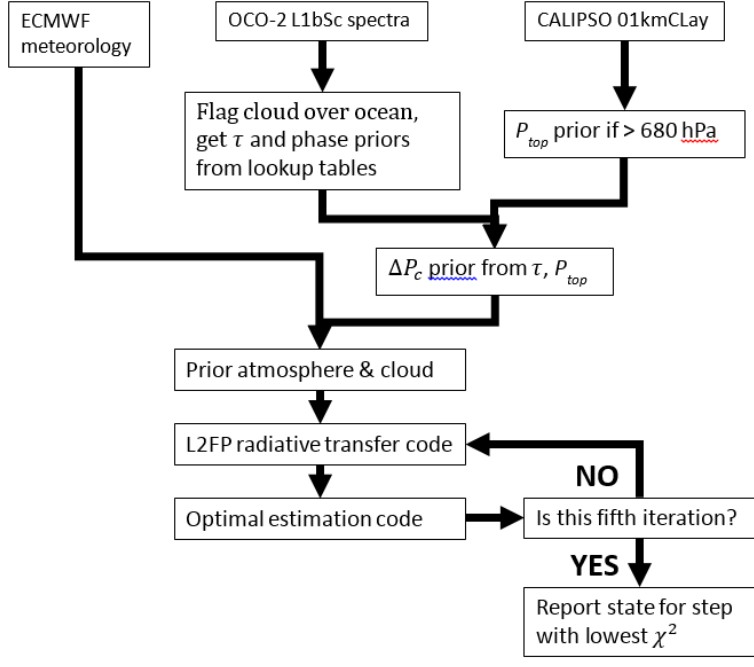

**Figure 3. Flow diagram illustrating the OCO2CLD-LIDAR-AUX retrieval.**

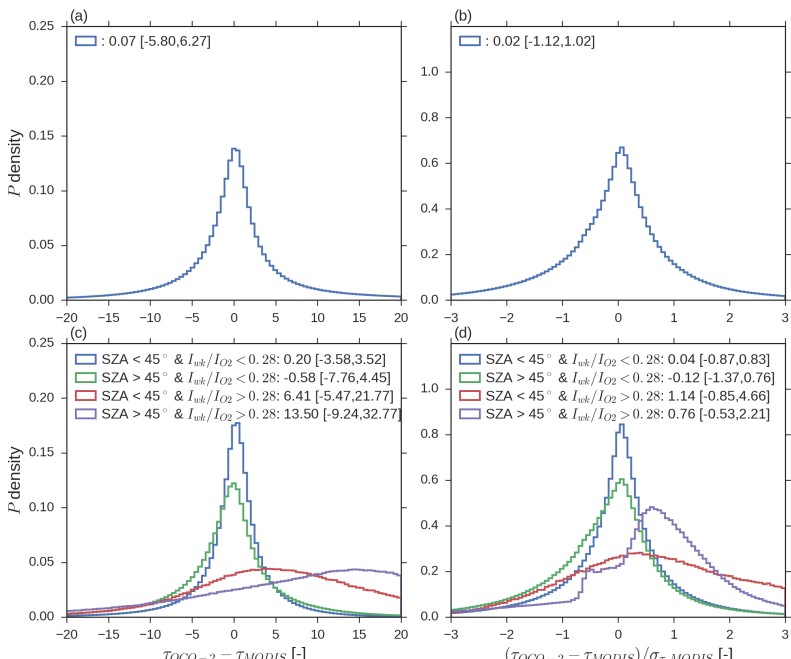

**Figure 4. OCO-2 minus MODIS $\tau$ statistics, in all cases the legend reports the median [14th, 86th percentiles]. (a) all retrievals absolute differences. (b) as (a) but each difference is divided by the matching MODIS uncertainty. (c) retrievals split by SZA and radiance ratio warn flag, and (d) as (c) but divided by MODIS uncertainties as in (b). These separations are part of the OCO2CLD-LIDAR-AUX Quality_flag: SZA above 45° adds 1 to Quality_flag, and $I_{wk}/I_{O2}$ above 0.28 adds 2.**

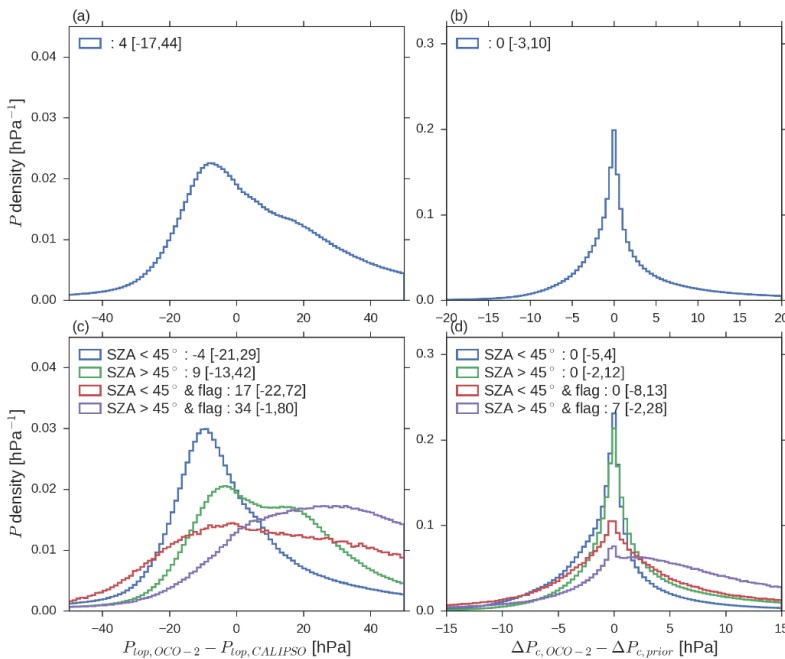

**Figure 5. Posterior minus prior retrieval statistics, all legends report median [14th, 86th percentiles]. (a) all OCO-2 minus CALIPSO cloud top pressure, (b) all OCO-2 posterior minus prior cloud pressure thicknesses, (c) as (a) but subset by Quality_flag, (d) as (b) but subset by Quality_flag.**

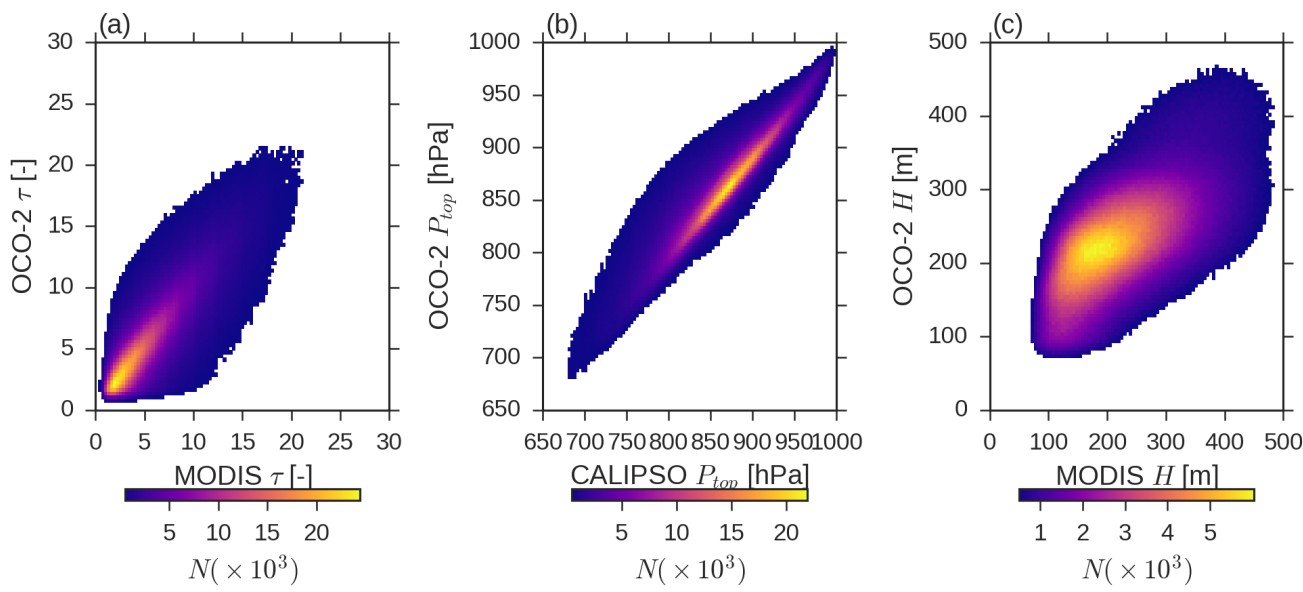

**Figure 6. 2d histograms of (a) MODIS versus OCO-2 cloud $\tau$ (b) CALIPSO versus OCO-2 $P_{top}$ and (c) MODIS versus OCO-2 $H$, with MODIS $H$ being equivalent thickness based on retrieved liquid water path combined with Equation (4) and $f_{ad}$ = 1. Bins with N < 500 are masked and only OCO2CLD-LIDAR-AUX Quality_flag = 0 retrievals are shown.**

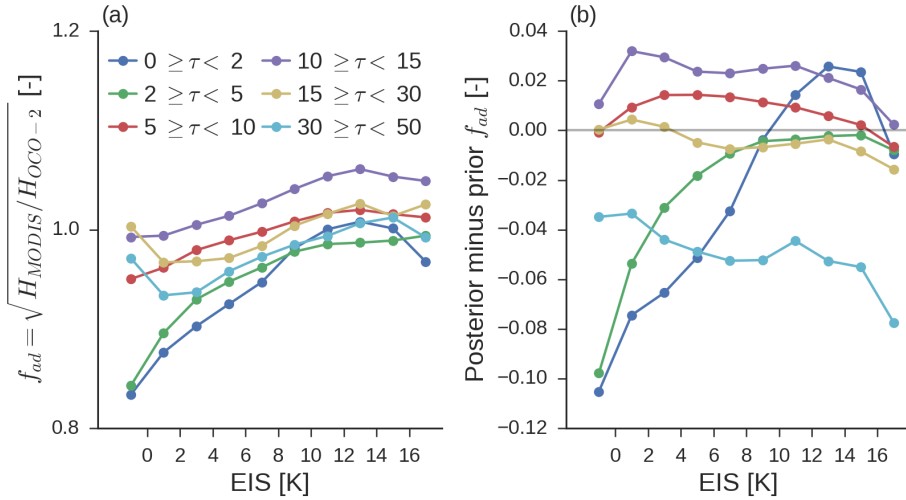

**Figure 7.** Estimated adiabatic fraction of OCO-2 Quality_flag = 0 retrieved clouds relative to MODIS, assuming equal liquid water paths and binned by estimated inversion strength. The first bin includes all cases where EIS < 0 °C and the last bin includes all cases where EIS > 16 °C. (a) posterior $f_{ad}$ split by cloud $\tau$, (b) the change in OCO-2 $f_{ad}$ as posterior minus prior.

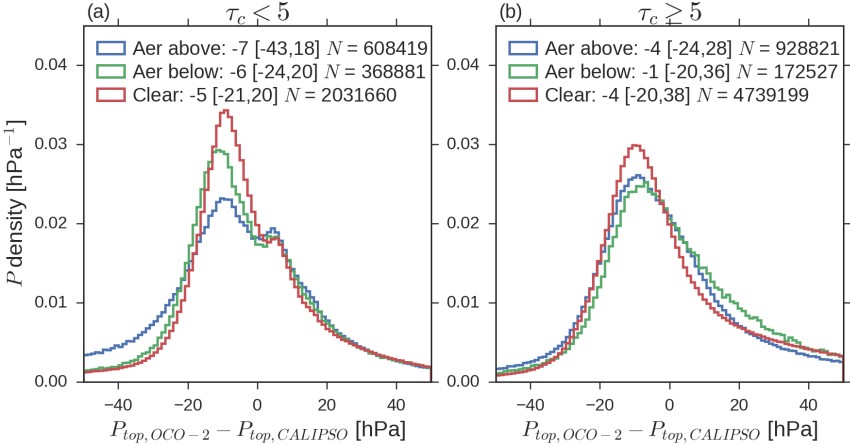

**Figure 8.** OCO-2 minus CALIPSO $P_{top}$ statistics for Quality_flag = 0 cases, subset for aerosol layer above cloud (N~1.5×10⁶), aerosol layer below cloud (N~0.5×10⁶) or no aerosol layer detected (N~6.8×10⁶) by CALIPSO's 05kmALay. Legend reports median [14th, 18th percentiles]. (a) clouds with posterior $\tau < 5$, (b) clouds with posterior $\tau \geq 5$.

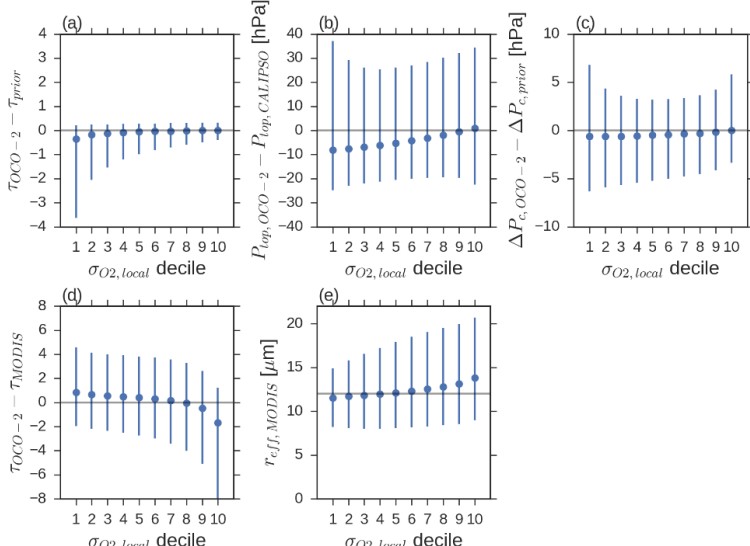

**Figure 9. Statistics of Quality_flag = 0 cloud cases split into deciles based on the standard deviation of continuum A-band radiances in neighbouring footprints divided by the footprint's continuum. The points are the bin medians and the bars cover the 14th—86th percentiles. (a) OCO-2 posterior minus prior $\tau$, (b) OCO-2 minus CALIPSO $P_{top}$, (c) OCO-2 posterior minus prior $\Delta P_c$, (d) OCO-2 minus collocated MODIS $\tau$, (e) collocated MODIS $r_e$. The horizontal line in (e) is at $r_{e,h} = 12\ \mu m$ to indicate the OCO2CLD-LIDAR-AUX assumed value.**

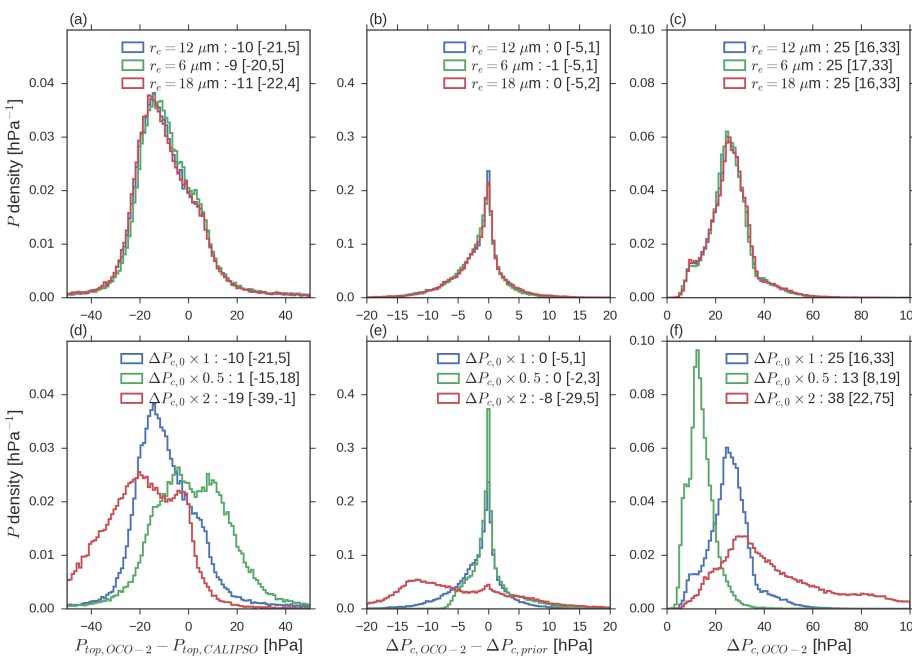

**Figure 10. Retrieval properties for 10 orbits when assumed $r_{e,h}$ and prior $\Delta P_c$ are changed. (a) OCO-2 minus CALIPSO $P_{top}$ difference for default $r_{e,h} = 12\ \mu m$ compared with $6\ \mu m$ and $18\ \mu m$ (b) $\Delta P_c$ difference for changed $r_{e,h}$, (c) $\Delta P_c$ retrieved for changed $r_e$, (d) $P_{top}$ difference for default $\Delta P_c$ prior, and for when this is scaled by 0.5 or 2.0, (e) $\Delta P_c$ for the same, (f) retrieved $\Delta P_c$ for the same.**

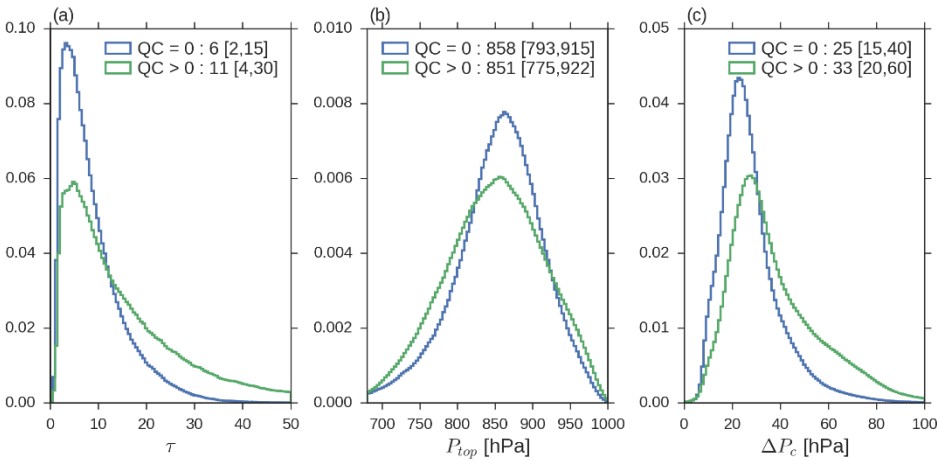

**Figure 11. Distributions of retrieved cloud properties over full OCO2CLD-LIDAR-AUX dataset, split by Quality_flag = 0 (QC = 0 in legend) or Quality_flag > 0. (a) cloud optical depth, (b) cloud top pressure, (c) cloud pressure thickness.**

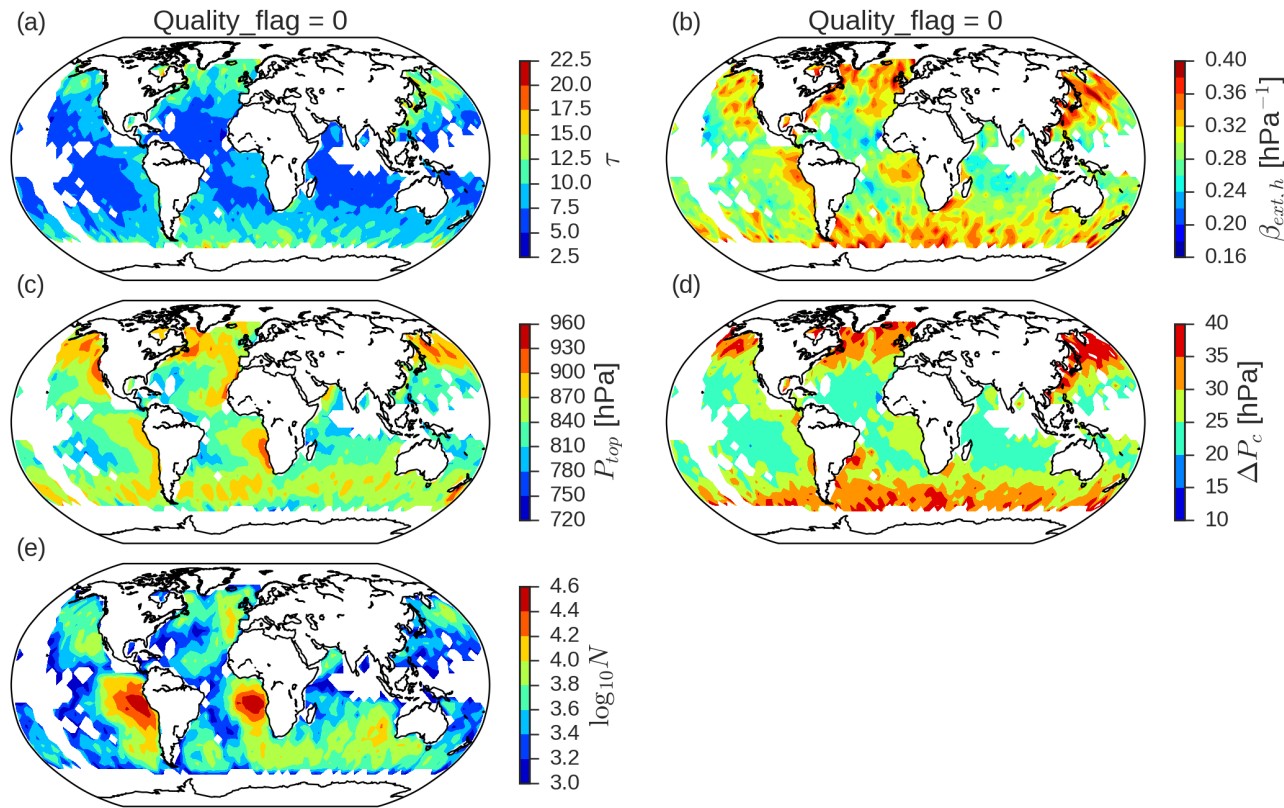

**Figure 12. Mean retrieved properties over full OCO2CLD-LIDAR-AUX record on a 4°×4° latitude-longitude grid, only Quality_flag = 0 retrievals are included. (a) Cloud optical depth, (b) homogeneous cloud extinction coefficient $\beta_{ext,h} = \tau/\Delta P_c$, (c) cloud top pressure, (d) cloud pressure thickness, (e) Logarithm of retrieval count.**

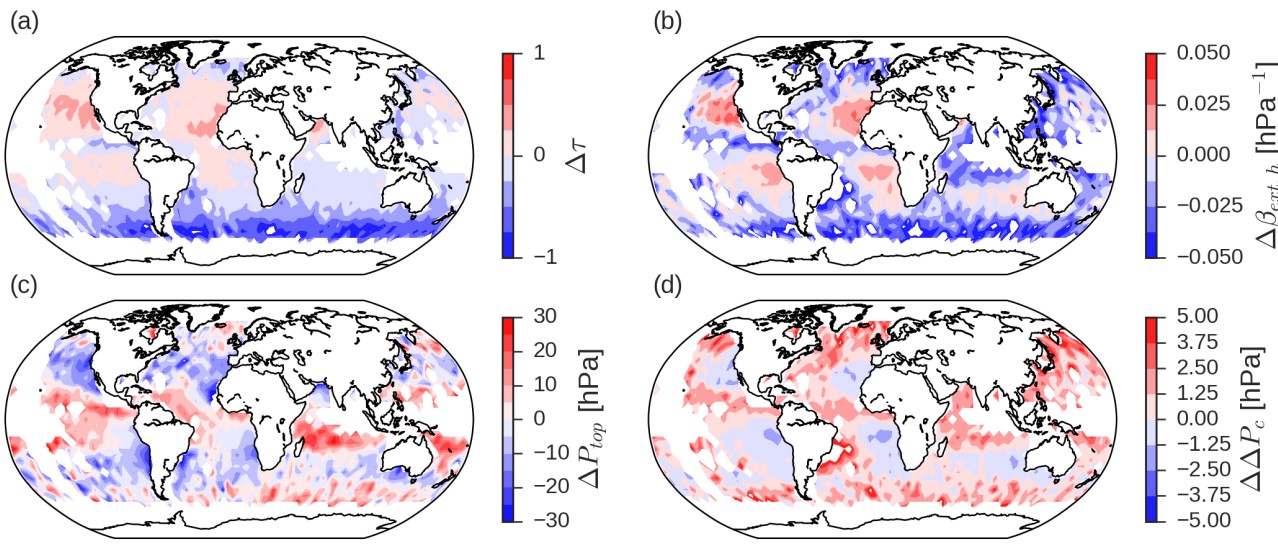

**Figure 13. Changes in retrieved properties as posterior minus prior, for Quality_flag = 0 retrievals, mean value in each 4°×4° latitude-longitude grid cell.**

**Table 1. Construction of each input vector or matrix in retrieval.**

| Property | Long name | Description |
|---|---|---|
| $\mathbf{x}_{a,1}$ | Prior $\ln \tau$ | A-band continuum lookup table from Richardson et al. (2017) |
| $\mathbf{x}_{a,2}$ | Prior $\ln P_{top}$ | CALIPSO 01kmCLay value |
| $\mathbf{x}_{a,3}$ | Prior $\ln \Delta P_c$ | Equation (1) with $r_{e,h}$ = 12 $\mu$m |
| $\mathbf{S}_{a,1,1}$ | Prior $\ln \tau$ covariance | $0.20^2$, equivalent to $\pm 20$ % in $\tau$ |
| $\mathbf{S}_{a,2,2}$ | Prior $\ln P_{top}$ covariance | $(5/P_{top})^2$, equivalent to $\pm 5$ hPa in $P_{top}$ |
| $\mathbf{S}_{a,3,3}$ | Prior $\ln \Delta P_c$ | $0.25^2$, equivalent to $\pm 25$ % in $\Delta P_c$ |
| $\mathbf{S}_{a,i \neq j}$ | Off-diagonal prior covariance | 0 |
| $\mathbf{K}_{1,:}$ | $\ln \tau$ Jacobian | Forward model finite difference with $\delta\tau$=0.01 |
| $\mathbf{K}_{2,:}$ | $\ln P_{top}$ Jacobian | Forward model finite difference with $\delta P_{top}$=0.1 hPa |
| $\mathbf{K}_{3,:}$ | $\ln \Delta P_c$ Jacobian | Forward model finite difference with $\delta\Delta P_c$ = 0.1 hPa |
| $\mathbf{S}_{\epsilon}$ | Observation covariance | Measurement SNR plus scaled pre-computed values from Richardson and Stephens (2018) |

**Table 2. Product names, version numbers and citations for key inputs and the forward model used in OCO2CLD-LIDAR-AUX**

| Retrieval input | Description |
|---|---|
| **Spacecraft location & view geometry** | OCO-2 L1bSc Version 7 |
| **OCO-2 spectra and SNR** | OCO-2 L1bSc Version 7 |
| **CALIPSO $P_{top}$** | 01kmCLay Version 4 collocated with CALIPSO as described in (Taylor et al., 2016) |
| **Meteorology** | ECMWF forecast interpolated onto OCO-2 footprints, as described in OCO-2 Version 7 ATBD (Boesch et al., 2017) |
| **Spectroscopy** | OCO-2 Absorption Coefficient (ABSCO) tables version 5 (Drouin et al., 2016) |
| **Radiative transfer forward model** | Level 2 Full Physics as in OCO-2 Version 7 release, modified as described in (Richardson et al., 2017). Latest version at: https://github.com/nasa/RtRetrievalFramework |

**Table 3. Components of Quality_flag. If a retrieval is summed, then the final Quality_flag is the sum of all of these Quality_flag values.**

| Quality_flag value | Meaning |
|---|---|
| -999999 | No retrieval attempted |
| 0 | Retrieval successful with no warnings |
| 1 | SZA > 45° |
| 2 | Low $I_{wk}/I_{O2}$ ratio, risk of poor retrieval |
| 4 | Cosmic ray strike on detector |
| 8 | Retrieved state outside recommended range |
| 32 | Code failure |

**Table 4. Statistics of a test sample of OCO2CLD-LIDAR output**

| | |
|---|---:|
| Number of orbits | 3162 |
| Attempts per orbit | 7336.4 |
| Success percent | 93.69 |
| Cosmic ray strike percent | 0.12 |
| Outside retrieval space | 4.56 |
| Code fail | 2.01 |
| OCO-2 success + MODIS liquid | 84.48 |
| OCO-2 success + MODIS liquid tau retrieved | 70.97 |