# Peer review of "Marine liquid cloud geometric thickness retrieved from OCO-2's oxygen A-band spectrometer"

_Atmospheric Measurement Techniques, 2018_

## Referee Comment (RC1) · Anonymous Referee #3 · 25 Nov 2018

A single-layered cloud optical thickness, pressure thickness and cloud- top pressure retrieval based on oxygen A-band radiance measurements is certainly a worthwhile endeavor and has been a long time coming.

Recommendation: Accept for publication after minor corrections.

Comments:

1) The discussion on Lines 20-32 of Page 3 is important and needs to be crystal clear.

When stating that clouds are homogeneous, make sure to always make clear when you mean vertically homogeneous versus horizontally homogeneous. On Line 26 you write

"homogeneous plane-parallel clouds" and I originally took this to mean "horizontally homogeneous plane-parallel clouds" but then I came to think that you meant "vertically homogeneous clouds that are also horizontally homogeneous, plane-parallel clouds".

Is the effective radius re on Line 20 of Page 3 the same as the effective radius r,ad,top on Line 29 of Page 3? And is the effecive radius reff on Line 9 of Page 4 for MODIS really a vertically homogeneous cloud effective radius so perhaps the same as re,h on Line 29 of Page 3? Anything that is done to tighten up the meaning of symbols would be helpful in this part of the manuscript.

2) Your horizontal spatial variability parameter (Sect. 5.3.2) will capture some variability but perhaps not some of the important variability as you state. Would it be worthwhile to manually identify scenes where gaps of various sizes exist within a single-layered cloud deck and determine whether or not the retrieval is influenced by ever longer paths through the gaps in such single-layered cloud decks?

3) Lines 14-21 on Page 4 are satellite centric and ignore all of the ground-based measurements that are relevant to this problem. Wouldn't ARM MAGIC measurements be of value here?

4) The words "Direct measurement" in the Section 2.2 heading and "more direct" on Line 23 thereafter seem off target. You are retrieving cloud thickness not measuring it and the retrieval is anything but direct, as all of the words in the pages to come indicate.

5) You need references at the end of Line 3 on Page 5: so "developed (References?)." I would recommend expanding the references to ground-based research too and Qilong Min is one person who pursued oxygen A-band retrievals using ARM data about 10 years back now.

6) Lines 16-19 on Page 5: Qilong Min also investigated the spectral width necessary for oxygen A-band retrievals of cloud structure.

Minor Details:

1) Perhaps the title should start as "Marine liquid-cloud ..." instead of "Liquid marine cloud..."?

2) The measurements are the radiances, correct? Everything else is retrieved, correct? If so, be sure to use the words "measurements" and "retrievals" to reflect this fact. Line 16 of the abstract: "Measurements are of ..." should be "Retrievals are of ...", correct? This occurs a lot throughout the paper.

3) There are minor nonsense phrases throughout the paper because of typos and omissions. An example is Line 4 on Page 15:

"which we Section 5.3.3 linked to ..."

which is much more understandable when the parentheses are added:

"which we (Section 5.3.3) linked to ..."

Other examples are

Line 6, Page 2: "measured by other satellite products" Line 15, Page 12: "This product averages along"

Products do not measure nor average so what do these phrases really mean? Fixing these types of errors throughout the manuscript would further improve its readability.

———————————————

---

## Referee Comment (RC2) · Anonymous Referee #1 · 18 Dec 2018

This manuscript presents a method for the joint retrieval of cloud optical thickness, top pressure and geometric thickness from passive hyperspectral shortwave IR measurements combined with lidar observations. Overall the paper is well written and the presented methodology appears sound. The retrieval of the cloud geometric thickness from $O_2$ A-band measurements is an important novel element of this method, even though it must be acknowledged that the validation of cloud geometric thickness looks much less robust than that of the other parameters. In particular, I find it difficult to understand what is the additional added value of the retrieved cloud geometric thickness compared to a reasonably chosen prior. Below are my complete comments.

[Figure]

MAIN COMMENTS

- The validation of the retrieved quantities is mostly focused on discussing biases and their most likely sources, but it would be also nice to see how well do the retrieved cloud optical thickness correlate with MODIS.

- While correlative data are available for the validation of cloud optical thickness (MODIS) and cloud top pressure (CALIPSO), the only available verification for cloud geometric thickness is a comparison to the adiabatic prior. I know that independent measurements of cloud geometric thickness are difficult to obtain, but nevertheless don't you feel that just comparing the retrieval with the prior limits somehow our capability of assessing the added value of the retrieval? After all, if you carry out a retrieval it is because you would like to get better estimates than the prior. What I see from the paper is that your Delta P_c retrievals are sensitive to the choice of the prior. As you say at page 13, scaling the prior Delta P_c by 0.5 or 2 leads to posterior Delta P_c that are, in your own opinion, unrealistically small and unrealistically large respectively. Absent a dataset of independent measurements, though, it is difficult to corroborate this opinion. Would an experiment with synthetic data (running the retrieval on synthetic cloud scenes of which you know the geometric thickness) be of any help? And are there any instruments (e.g., ground-based lidars) available from which marine cloud base heights can be determined and combined with CALIPSO cloud top heights? Wouldn't that help you understand more precisely how your Delta P_c retrievals behave?

MINOR COMMENTS

- P4, L14. Try to avoid the repetition in "A term-by-term error analysis estimated H could be estimated..." (replace one of the two "estimated" with a synonym)

- P7, L7. "each I" -> "each I_i (i=1,...,75)"

- I think the description of the optimal estimation principles at page 7 is a bit too terse, and may not be help a non-expert reader to understand what this all is about. No

context is provided for the invocation of Bayes' theorem. In my opinion, the following points should emerge from the description:

1. It is assumed that the state vector follows a Gaussian distribution with mean $x\_a$ and covariance matrix $S\_a$

2. It is assumed that the measurement error is additive and follows a Gaussian distribution with zero mean and covariance matrix $S\_epsilon$

3. The Bayes theorem is applied in order to express the posterior probability density of the state vector given the observations

4. The estimation procedure looks for the maximum of such probability density

- P7, L16. Please clarify that $x\_i$ is the iterate solution, and that $K\_i$ is the Jacobian matrix of the forward model evaluated at $x\_i$

- P7, L20. What convergence criterion did you adopt?

- P8, L31. Please specify what you mean by "correct for $mu\_0$". I guess you divide I by $mu\_0$, but it would be better to make this explicit.

- P9, L4. Could you explain the reason for applying a low cloud top pressure threshold? Is it because you assume ice clouds if $p\_top < 680$ hPa?

- P11, L1. I may have missed where $I\_wk$ and $I\_O2$ are defined. If they weren't, please specify their meaning (I guess they represent radiances in the weak CO2 band and in the O2-A band respectively, but it should be made explicit).

- P11, L8-20. I had some difficulties trying to link the text with what is shown in Fig. 4. The subfigures (c) and (d) contain four plots each. In the legend, two plots are marked with "& flag" and two are not. Does the "& flag" mark mean $I\_wk/I\_O2<0.28$? It would be handy to have this information readily available in the figure caption (now you only mention a "radiance ratio warn flag"). Furthermore, at L13 you say that "for the full sample the median bias is 0.02 times the MODIS uncertainty and the 14-86% range is

-1.15 to 0.99". Are you then referring to panel (b) of Figure 4? If so, there the 14-86% range reads -1.12 to 1.02. A similar question holds for the last sentence (L18-20). Are you referring to panel (d) of Fig. 4? If so, the numbers mentioned in the text seem slightly inconsistent with those reported in the figure. This is of course a minor issue, but it does not help readability.

- P15, L4. "we Section 5.3.3 linked" -> "in Section 5.3.3 we linked" ?

- Wouldn't the material presented as supplement be more suitable as an appendix inside the manuscript?

---

## Referee Comment (RC3) · Anonymous Referee #2 · 23 Dec 2018

Review comments on manuscript "Liquid marine cloud geometric thickness retrieved from OCO-2's oxygen A-band spectrometer"

Authors: M. Richardson et al. MS No.: amt-2018-387 MS Type: Research article

General comments:

This paper introduces the new OCO2CLD-LIDAR-AUX product and its algorithm theoretical basis. The algorithm adopts the optimum estimation principles to retrieve cloud properties, including optical thickness, cloud top, and geometrical thickness for marine boundary layer clouds using the OCO-2 hyperspectral A-band measurements. Performance evaluation is also conducted. The paper is well written and this new product

provides new information for further understanding the properties of marine boundary layer clouds. The topic is suitable for publication in AMT. I recommend publication after some minor revisions. Some concerns for the authors to consider:

1) I actually don't find the physics of cloud phase detection method used here straight-forward. What is the general value range for Iwk/Io2? How strongly does it depend on other factors in addition to cloud phase (e.g., cloud optical depth, height etc)? 2) The differences in performance for thin and thick clouds (Fig. 6) makes me wonder the role of surface reflectance. How is the sea surface reflectance handled?

Specific comments:

P8 L19: It is mentioned that the L2RTM input includes meteorology. I assume this include temperature profile?

P9 L8: Cloud phase determination is brought up here, but the details are given in P11; suggest either move the details here or add something like "detailed discussion in Section5.1".

Figure 4: Does "&flag" in the legend mean "Quality_flag =2"?

Figure 8: there are typos in the caption: there are two panel "c" descriptions (the second should be for panel d) but none for panel "f".
* * *

---

## Author Comment (AC1) · 12 Feb 2019

A single-layered cloud optical thickness, pressure thickness and cloud- top pressure retrieval based on oxygen A-band radiance measurements is certainly a worthwhile endeavor and has been a long time coming.

Recommendation: Accept for publication after minor corrections.

**Comments:** We appreciate your time and consideration. And we are happy that you recommend publication following minor corrections. We believe we have addressed all of your concerns below, and a version of the manuscript with changes tracked follows this review. In particular, you have helped us to make the language tighter and more precise, thank you.

**Changes:** See below

Comments:

1) The discussion on Lines 20-32 of Page 3 is important and needs to be crystal clear. When stating that clouds are homogeneous, make sure to always make clear when you mean vertically homogeneous versus horizontally homogeneous. On Line 26 you write "homogeneous plane-parallel clouds" and I originally took this to mean "horizontally homogeneous plane-parallel clouds" but then I came to think that you meant "vertically homogeneous clouds that are also horizontally homogeneous, plane-parallel clouds". Is the effective radius re on Line 20 of Page 3 the same as the effective radius $r_{ad,top}$ on Line 29 of Page 3? And is the effecive radius reff on Line 9 of Page 4 for MODIS really a vertically homogeneous cloud effective radius so perhaps the same as $r_{e,h}$ on Line 29 of Page 3? Anything that is done to tighten up the meaning of symbols would be helpful in this part of the manuscript.

**Comments:** We really have 5 effective radii: (i) cloud-top, (ii) that which gives the same optical depth in a vertically homogeneous cloud with the same LWP, H and $\tau$ (5/6ths of top for non-absorbing channels), (iii) the value used in our retrievals, (iv) the value retrieved by MODIS, and (v) the value at some given height. The original manuscript was not clear enough, but we think we have found a better system.

**Changes:** Many changes throughout, including:

1) P3 Text now reads: "… it is common for cloudy radiative transfer to assume plane-parallel clouds that are both horizontally and vertically homogeneous" and "vertically" has been inserted before "homogeneous" in the following lines."

2) Eq. (1) and its description "$r_e$" → "$r_{ead_2}$"

3) MODIS $r_{eff}$ is now labelled "$r_{eM}$"

4) Old P4 L8 MODIS description has added text: "Simulations by Platnick (2000) suggest that the retrieved $r_{eM}$ is smaller than $r_{ead_2}$ as the channel weighting functions are below cloud top, but that the ratio depends on the MODIS channel used, $r_e$ profile and somewhat on the cloud optical depth. If the MODIS retrieval performs similarly to those simulations, then $r_{eM}$ is similar to $r_{eh}$ according to the results in Platnick (2000) Table 3a"

5) Figure 1 caption text changed to: "(a) simulated A-band reflectance spectrum for a vertically homogeneous cloud with $\tau = 10$, $r_{eh} = 12$ μm,"

6) Section 4.2 text includes changes with $r_{eM}$, $r_{ead_2}$ and $r_{eh}$ used where we feel appropriate. In some cases we leave it as "$r_e$" when a qualifying adjective beforehand specifies the meaning. The text also now refers to Section 2.1 which includes the changes from our point 4) above.

2) Your horizontal spatial variability parameter (Sect. 5.3.2) will capture some variability but perhaps not some of the important variability as you state. Would it be worthwhile to manually identify scenes where gaps of various sizes exist within a single-layered cloud deck and determine whether or not the retrieval is influenced by ever longer paths through the gaps in such single-layered cloud decks?

**Comments:** We decided not to do further analysis on this because:

1) Our selected metric should correlate both with within- and between- footprint variability in the cloud field. We expect to capture gaps using our radiance metric, and since we're over a dark surface the 3d radiative effects associated with a gap of ~2 km (one footprint gap) should be similar to those of 4+ km (>1 footprint gap)

2) Defining a gap is somewhat tricky. We could use our flags, or MODIS flags, and also need to exclude aerosol and ice clouds

3) Manually selecting scenes by loading MODIS imagery is time consuming for even a small sample size

We plan to better investigate nonuniform cloud fields in future but feel the extra effort required at this point does not add to our findings.

**Changes:** N/A

3) Lines 14-21 on Page 4 are satellite centric and ignore all of the ground-based measurements that are relevant to this problem. Wouldn't ARM MAGIC measurements be of value here?

Comments: We have added paragraphs here to discuss this and explain why we don't use ARM Azores or Ascension (too far away) or MAGIC (measurements finished before OCO-2 launch) for validation. We are working on further tests of the thickness retrieval.

Changes: Paragraph added to discuss surface measurements including MAGIC and ARM sites on the Azores & Ascension.

4) The words "Direct measurement" in the Section 2.2 heading and "more direct" on Line 23 thereafter seem off target. You are retrieving cloud thickness not measuring it and the retrieval is anything but direct, as all of the words in the pages to come indicate.

Comments: Agreed.

Changes: Changes made:

2.2 Title: "Direct measurement" → "Explicit retrieval"

P4 L23: "An alternative, and more direct approach" → "An alternative approach"

P16: "…provide a physical independent method of obtaining thickness information" (deleted "that is direct")

5) You need references at the end of Line 3 on Page 5: so "developed (References?)." I would recommend expanding the references to ground-based research too and Qilong Min is one person who pursued oxygen A-band retrievals using ARM data about 10 years back now.

Comments:

Changes: Citations added: "…has been developed (Li and Min, 2010; Min et al., 2004)."

6) Lines 16-19 on Page 5: Qilong Min also investigated the spectral width necessary for oxygen A-band retrievals of cloud structure.

Comments: I originally skipped this due to its focus on very thin layers, but think you're right and that it adds to the discussion.

Changes: New text (underlined added): "These suggested that a spectral sampling of $0.5-1.0$ cm$^1$ is necessary for a joint retrieval, similar to the 0.5 cm$^1$ that Min and Harrison (2004) estimated as necessary to obtain four pieces of information in an atmosphere with optically thin scattering layers. These results are dependent somewhat on other instrument characteristics such as the signal-to-noise ratio."

Minor Details:

1) Perhaps the title should start as "Marine liquid-cloud ..." instead of "Liquid marine cloud..."?

Comments:

Changes: Title changed.

2) The measurements are the radiances, correct? Everything else is retrieved, correct? If so, be sure to use the words "measurements" and "retrievals" to reflect this fact. Line 16 of the abstract: "Measurements are of ..." should be "Retrievals are of ...", correct? This occurs a lot throughout the paper.

**Comments:** We agree with this discrimination and have made changes where we feel appropriate, although not in the abstract because we feel that "measurements are of single-layer clouds" is an accurate description since the measured radiances we use *are* of single layer clouds, even though we then do something (i.e. the retrieval) *to* these measurements after.

**Changes:** Examples of changes made (with replacement for "measurement" or similar underlined):

1) Section 1:"(OCO-2) primary mission is to retrieve"
2) P4 "due to the difficulty of retrieving $H$ of these clouds"
3) P4 "…demonstrated using retrievals based on combined measurements…"
4) P12 "shows that the OCO-2 retrievals result in increased cloud altitude…"
5) P13 "This is not a direct estimate of the within-footprint variability…"
6) P15: "in agreement with estimates from field measurements"
7) P16: "Hyperspectral A-band retrievals are based on photon path length…"

3) There are minor nonsense phrases throughout the paper because of typos and omissions. An example is Line 4 on Page 15: "which we Section 5.3.3 linked to ..." which is much more understandable when the parentheses are added: "which we (Section 5.3.3) linked to ..."

**Comments:** Thanks for the catch

**Changes:** Parentheses added.

Other examples are

Line 6, Page 2: "measured by other satellite products" Line 15, Page 12: "This product averages along" Products do not measure nor average so what do these phrases really mean? Fixing these types of errors throughout the manuscript would further improve its readability.

**Comments:**

**Changes:** Changes made (referring to original page/line numbers):

P2 L6L: "$\Delta P_c$ is poorly measured by other satellite products" → "$\Delta P_c$ is poorly constrained by other satellite products"

[revised manuscript text omitted]

---

## Author Comment (AC2) · 12 Feb 2019

This manuscript presents a method for the joint retrieval of cloud optical thickness, top pressure and geometric thickness from passive hyperspectral shortwave IR measurements combined with lidar observations. Overall the paper is well written and the presented methodology appears sound. The retrieval of the cloud geometric thickness from O2 A-band measurements is an important novel element of this method, even though it must be acknowledged that the validation of cloud geometric thickness looks much less robust than that of the other parameters. In particular, I find it difficult to understand what is the additional added value of the retrieved cloud geometric thickness compared to a reasonably chosen prior. Below are my complete comments.

**Comments:** Thank you for taking the time to review our paper. We completely agree that it is hard to test the robustness of our $H$ retrievals, and suspect there are biases in it. We have worked on the phrasing to keep it clear that this research project requires further work and added two new figures, one showing indirect evidence of useful new information in our retrieved $H$ and another showing comparison versus MODIS-implied $H$.

We agree with your overall point that our conclusions regarding $H$ are tentative but think we show a serious advance, being the first hyperspectral A-band $H$ retrieval attempted from space.

We are confident that we have addressed concerns that would prevent publication, but have been careful in our language so that readers can understand the limitations and future work required. A full redlined version of the manuscript is in the file submitted in response to RC1.

**Changes:** Please see individual responses below

**MAIN COMMENTS**

- The validation of the retrieved quantities is mostly focused on discussing biases and their most likely sources, but it would be also nice to see how well do the retrieved cloud optical thickness correlate with MODIS.

**Comments:**

**Changes:** We have added a new Figure 6 that shows retrieved $\tau$ veruss MODIS, ctP versus CALIPSO and $H$ versus that implied from MODIS-retrieved LWP, using ECMWF profiles to derive $c_w$ and assuming

- While correlative data are available for the validation of cloud optical thickness (MODIS) and cloud top pressure (CALIPSO), the only available verification for cloud geometric thickness is a comparison to the adiabatic prior. I know that independent measurements of cloud geometric thickness are difficult to obtain, but nevertheless don't you feel that just comparing the retrieval with the prior limits somehow our capability of assessing the added value of the retrieval? After all, if you carry out a retrieval it is because you would like to get better estimates than the prior. What I see from the paper is that your Delta P_c retrievals are sensitive to the choice of the prior. As you say at page 13, scaling the prior Delta P_c by 0.5 or 2 leads to posterior Delta P_c that are, in your own opinion, unrealistically small and unrealistically large respectively. Absent a dataset of independent measurements, though, it is difficult to corroborate this opinion. Would an experiment with synthetic data (running the retrieval on synthetic cloud scenes of which you know the geometric thickness) be of any help? And are there any instruments (e.g., ground-based lidars) available from which marine cloud base heights can be determined and combined with CALIPSO cloud top heights? Wouldn't that help you understand more precisely how your Delta P_c retrievals behave?

**Comments:** Richardson & Stephens (2018) shows a synthetic retrieval and error statistics, but the "true" clouds were vertically homogeneous too. Future work will test nonuniform vertical cloud profiles, a new wrapper for the RT code reduces difficulties related to cloudy profiles and $r_{ef}$, and should be working soon.

We couldn't find surface validation data but present indirect evidence for new & useful information. We binned $H$ by both $\tau$ and estimated inversion strength from ECMWF then compute an adiabatic fraction relative to the $H$ implied by MODIS-derived $LWP$. We see that our retrieved $H$ is consistent with less-adiabatic conditions when EIS is weaker, although it's only at lower $\tau$ values where the retrieval greatly updates the prior as expected. Importantly, these changes are independent of and represent information independent of that from MODIS.

If our retrievals are good, we would expect this pattern of H in response to EIS. But observing this pattern obviously doesn't mean that our retrievals are good. So while we think this is promising evidence in favour of our retrieval approach being useful, we have tried to be clear in our phrasing that these results are tentative and are a first of a kind.

**Changes:**
- Text added to Section 2.1 to discuss validation data availability, including surface validation
- Text added to Section 5.2 and its title changed to "Cloud top pressure versus CALIPSO, geometric thickness versus MODIS and implied subadiabaticity"
- New Figure 7 added, showing changes in implied adiabaticity with EIS
- Modifications to Section 7 to summarise our evidence, new text: "Nevertheless, the small discrepancies relative to MODIS optical depth (Figure 4), the tendency to retrieve more subadiabatic clouds under weaker inversions, at least for optically thinner clouds (Figure 7), and the increased extinction coefficient in the marine stratocumulus regions (Figure 13) suggest that the OCO-2 spectra add useful information"

MINOR COMMENTS

- P4, L14. Try to avoid the repetition in "A term-by-term error analysis estimated H could be estimated..." (replace one of the two "estimated" with a synonym)

**Comments:**

**Changes:** Changed to "error analysis suggested that H could be estimated to…"

- P7, L7. "each I" -> "each I_i (i=1,...,75)"

**Comments:**

**Changes:** Change made.

- I think the description of the optimal estimation principles at page 7 is a bit too terse, and may not be help a non-expert reader to understand what this all is about. No context is provided for the invocation of Bayes' theorem. In my opinion, the following points should emerge from the description:

1. It is assumed that the state vector follows a Gaussian distribution with mean x_a and covariance matrix S_a

2. It is assumed that the measurement error is additive and follows a Gaussian distribution with zero mean and covariance matrix S_epsilon

3. The Bayes theorem is applied in order to express the posterior probability density of the state vector given the observations

4. The estimation procedure looks for the maximum of such probability density

**Comments:** You have persuaded us that these are important details. While we have tried to keep it short and sweet, we have added the requested details.

**Changes:** New introduction to Section 4.1:

"We begin with a prior cloud state vector whose components are Gaussian, represented by a mean state vector $x_a$ and covariance matrix $S_a$. Meanwhile the observational uncertainty is represented by a zero-mean Gaussian with covariance matrix $S_\epsilon$. Optimal estimation produces a maximised posterior probability density of the posterior state given both the prior state and the observations, with appropriate weighting for their relative uncertainties. In our case the individual contributions to observational uncertainties are assumed to add in quadrature such that $S_\epsilon$ is simply the sum of each term's covariance. The posterior is estimated by applying Bayes' theorem assuming a linear forward model encapsulated in the Jacobian matrix K whose elements are $K_{i,j} = \partial y_i / \partial x_j$. "

- P7, L16. Please clarify that x_i is the iterate solution, and that K_i is the Jacobian matrix of the forward model evaluated at x_i

**Comments:** We missed how this was confusing, thanks for the careful reading.

**Changes:** Change time index to 'n' from 'i', leaving 'i' to refer to channel indices for observations/Jacobian.

- P7, L20. What convergence criterion did you adopt?

**Comments:** We did not use a convergence criterion for awkward computational reasons. The way in which the cloud retrieval code was "bolted on" to the OCO-2 L2FP code involves running a repopulating files and configurating everything
10 every time the soundings to be done within a file changed. We found it easier to just run all of the soundings six times, so we did that and picked the lowest $\chi^2$ step. Future versions with the new RT wrapper will follow a more standard approach.

**Changes:** Text added: "No explicit convergence criterion was adopted: all retrievals that did not trigger computational problems are reported and users are provided with both the state estimate and the $\chi^2$ for the retrieval step used."

15 - P8, L31. Please specify what you mean by "correct for mu_0". I guess you divide I by mu_0, but it would be better to make this explicit.

**Comments:**

**Changes:** Changed to "..and divide by…"

20 - P9, L4. Could you explain the reason for applying a low cloud top pressure threshold? Is it because you assume ice clouds if p_top < 680 hPa?

**Comments:** Basically yes. There were a small number of these and we used this threshold to exclude potential egregious icy outliers in our liquid cloud retrieval.

**Changes:** Replace "whose $P_{top} > 680$ hPa. The CALIPSO $P_{top}$ threshold limits our sample to the low cloud threshold of the
25 International Satellite Cloud Climatology Project (Rossow and Schiffer, 1991) and helps to filter out non-liquid clouds."
→ "whose $P_{top} > 680$ hPa. The CALIPSO $P_{top}$ threshold limits our sample to the low cloud threshold of the International Satellite Cloud Climatology Project (Rossow and Schiffer, 1991) and helps to filter out non-liquid clouds."

- P11, L1. I may have missed where I_wk and I_O2 are defined. If they weren't, please specify their meaning (I guess they
30 represent radiances in the weak CO2 band and in the O2-A band respectively, but it should be made explicit).

**Comments:** This was an oversight

**Changes:** Section 4.3: "..(A-band $6\times10^{19}$, weak $CO_2$ band $1\times10^{19}$ photons m$^{-2}$ sr$^{-1}$ $\mu$m$^{-1}$)" → "(A-band $\mu_0 I_{O2} = 6\times10^{19}$, weak $CO_2$ band $\mu_0 I_{wk}$ $1\times10^{19}$ photons m$^{-2}$ s$^{-1}$ sr$^{-1}$ $\mu$m$^{-1}$)"

Section 5.1: "the ratio of $I_w/I_{o2}$" → "the ratio of the 10-channel continuum radiances in each band $I_w/I_{o2}$"

- P11, L8-20. I had some difficulties trying to link the text with what is shown in Fig. 4. The subfigures (c) and (d) contain four plots each. In the legend, two plots are marked with "& flag" and two are not. Does the "& flag" mark mean I_wk/I_O2<0.28? It would be handy to have this information readily available in the figure caption (now you only mention a "radiance ratio warn flag"). Furthermore, at L13 you say that "for the full sample the median bias is 0.02 times the MODIS uncertainty and the 14-86% range is -1.15 to 0.99". Are you then referring to panel (b) of Figure 4? If so, there the 14-86% range reads -1.12 to 1.02. A similar question holds for the last sentence (L18-20). Are you referring to panel (d) of Fig. 4? If so, the numbers mentioned in the text seem slightly inconsistent with those reported in the figure. This is of course a minor issue, but it does not help readability.

**Comments:** You're right, these all needed fixing.

**Changes:**

(underlines are additions):

- Figure 4(c,d) legend labels changed to replace "flag" with Iwk/IO2 > 0.28 and add Iwk/IO2 < 0.28 when no flag.

- "For the full sample, the median bias…" → "For the full sample in Figure 4(b), the median bias…"

- "For Quality_flag = 0, the median bias…" → "For Quality_flag = 0, where SZA < 45° and Iwk/IO2 < 0.28, Figure 4(d) shows that the median bias…"

- Discussion of Figure 4 numbers corrected.

- P15, L4. "we Section 5.3.3 linked" -> "in Section 5.3.3 we linked" ?

**Comments:**

**Changes:** Changed to: "we (Section 5.3.3) linked…"

- Wouldn't the material presented as supplement be more suitable as an appendix inside the manuscript?

**Comments:** We think that either work, but I (MR) prefer to keep this as SI. I don't see the benefit of switching to an appendix.

**Changes:** N/A

---

## Author Comment (AC3) · 12 Feb 2019

5 General comments:

This paper introduces the new OCO2CLD-LIDAR-AUX product and its algorithm theoretical basis. The algorithm adopts the optimum estimation principles to retrieve cloud properties, including optical thickness, cloud top, and geometrical thickness for marine boundary layer clouds using the OCO-2 hyperspectral A-band measurements. Performance evaluation is also conducted. The paper is well written and this new product provides new information for further understanding the

10 properties of marine boundary layer clouds. The topic is suitable for publication in AMT. I recommend publication after some minor revisions. Some concerns for the authors to consider:

**Comments:** Firstly, thanks for reading our paper, we appreciate that you clearly thought about relevant physics throughout and are gratified that you recommend publication after minor revisions. We have addressed your comments below, and the manuscript with tracked changes is part of the file in response to RC1.

15 **Changes:** See below.

1) I actually don't find the physics of cloud phase detection method used here straight-forward. What is the general value range for Iwk/Io2? How strongly does it depend on other factors in addition to cloud phase (e.g., cloud optical depth, height etc)?

20 **Comments:** We have flipped the order of Figures 2 and 3 and lengthened the description: it's a "traditional" Nakajima-King approach relying on how ice absorbs relatively more strongly for the longer wavelength band.

There is little sensitivity to cloud top pressure since we are using continuum bands with little above-cloud absorption. There are sensitivities to the droplet/crystal size, but these are summarized in the Nakajima & King reference, and the relationship to $\tau$ can be inferred from the lookup table that is the black line in the phase lookup table figure.

25

**Changes:** The final paragraph of Section 4.3 has been changed to:

"If both of these tests agree on a cloud, then the continuum A-band and weak $CO_2$ $\mu_0 I$ are used to estimate cloud phase via a lookup table that exploits how ice absorbs more strongly than water in the weak $CO_2$ band relative to the A-band (Nakajima and King, 1990). A lookup table is also used to estimate the prior cloud optical depth from the continuum A-band radiance,

30 since more optically thick clouds tend to be brighter, and Figure 2 shows both the phase and $\tau$ lookup tables."

Figures 2 and 3 order switched.

2) The differences in performance for thin and thick clouds (Fig. 6) makes me wonder the role of surface reflectance. How is the sea surface reflectance handled?

**Comments:** We use a Cox-Munk scheme with ECMWF surface winds, tests showed little response of the retrieval to scaling the albedo because the oceans are typically so dark in the nadir anyway, and any signal is dominated by clouds. We now comment more on this, and for further illustration: median clear-sky nadir view $\mu_0^{-1}I$ is about 4 W m-2 sr-1 micron-1. Our threshold for any cloud is 15 W m-2 sr-1 micron-1. Our changes below explain this.

**Changes:**

Text added to Section 4.3 when discussing meteorology:

"which provides temperature and humidity profiles along with surface wind speed for the Cox-Munk sea surface reflectance model (Cox and Munk, 1954)"

Text added to Section 4.3 when discussing cloud flags:

"This threshold is equivalent to just over 15 W m² sr¹ $\mu$m¹, compared with the median $\mu_0^{-1}I$ near 4 W m² sr¹ $\mu$m¹ in clear sky conditions, according to the OCO-2 A-band preprocessor."

Text added to Section 5.3.1 in discussion of Figure 6 (now Figure 8):

"This shift fits with aerosol layers above the cloud shortening photon path lengths, and is inconsistent with a dominant role for increased surface reflection in scenes with a low value of retrieved cloud $\tau$. The secondary peak near 0 hPa in low-$\tau$ clouds might be related to signals returning from the surface with a longer path length counteracting the upward shift, but these only represent a small fraction of the total retrievals."

Specific comments:

P8 L19: It is mentioned that the L2RTM input includes meteorology. I assume this include temperature profile?

**Comments:**

**Changes:** Text added to the same paragraph:

"which provides temperature and humidity profiles along with surface wind speed for the Cox-Munk sea surface reflectance model (Cox and Munk, 1954)"

P9 L8: Cloud phase determination is brought up here, but the details are given in P11; suggest either move the details here or add something like "detailed discussion in Section5.1".

**Comments:** We hope that our response to comment 1) covers this.

**Changes:** See above

5

Figure 4: Does "&flag" in the legend mean "Quality_flag =2"?

**Comments:**

**Changes:** We have changed the legend to explicitly state the ratio used and the final sentence of the caption has been lengthened to link these values to the Quality_flag.

10

Figure 8: there are typos in the caption: there are two panel "c" descriptions (the second should be for panel d) but none for panel "f".

**Comments:** Thanks for catching this.

**Changes:** Caption corrected.